



# Decreases in wintertime total column ozone over the Tibetan Plateau during 1979-2017

Yajuan Li[1,2], Martyn P. Chipperfield[2,3], Wuhu Feng[2,4], Sandip S. Dhomse[2,3], Richard J. Pope[2,3], Faquan Li[5], Dong Guo[6]

*1 School of Electronic Engineering, Nanjing Xiaozhuang University, Nanjing, China*
*2 School of Earth and Environment, University of Leeds, Leeds, UK*
*3 National Centre for Earth Observation, University of Leeds, Leeds, UK*
*4 National Centre for Atmospheric Science, University of Leeds, UK*
*5 Wuhan Institute of Physics and Mathematics, Chinese Academy of Sciences, Wuhan, China*
*6 Key Laboratory of Meteorological Disaster, Ministry of Education/Joint International Research Laboratory of Climate and Environment Change/Collaborative Innovation Center on Forecast and Evaluation of Meteorological Disasters, Nanjing University of Information Science & Technology, Nanjing, China*

**Abstract.** We use the ozone dataset from the Copernicus Climate Change Service (C3S) during 1979-2017 to investigate the long-term variations of the total column ozone (TCO) and the relative total ozone low (TOL) over the Tibetan Plateau (TP) during different seasons. Based on various regression models, the wintertime TCO over the TP decreases overall during 1979-2017 with ongoing decreases since 1997. We perform multivariate regression analysis to quantify the influence of dynamical and chemical processes responsible for the long-term TCO variability over the TP. We use both piecewise linear trend (PWLT) and equivalent effective stratospheric chlorine loading (EESC) -based regression models that include explanatory variables such as the 11-year solar cycle, quasi-biennial oscillation (QBO) at 30 hPa and 10 hPa and the geopotential height (GH) at 150 hPa. The 150 hPa GH is found to be a major dynamical contributor to the total ozone variability (8%) over the TP in wintertime. We also find strong correlation between TCO in DJF and the following JJA, indicating that negative/positive anomalies in the wintertime build up persist into summer. We also use the TOMCAT/SLIMCAT 3-D chemical transport model to investigate the contributions of different factors to the ozone variations over the TP. Using identical regression model on simulated TCO time series, we obtain consistent results with C3S-based data. We perform two sensitivity experiments with repeating dynamics of 2004 and 2008 to further study the role that the GH at 150 hPa plays in the ozone variations over the TP. The GH differences between the two years show an obvious, negative centre near 150 hPa over the TP in DJF. Composite analysis show that GH fluctuations associated with Inter Tropical Convergence Zone, ENSO events or Walker circulation play a key role in controlling TCO variability in the lower stratosphere.



## 1 Introduction

The Tibetan Plateau (TP), also known as the third pole, is one of the areas most sensitive to global climate change. It exerts important thermal and dynamical effects on the general circulation and climate change (Yanai et al., 1992; Ye and Wu, 1998). Furthermore, climate
changes over the TP have a significant impact on the distribution of stratospheric ozone. By acting as an important greenhouse gas and ultraviolet radiation absorber, variation of the ozone amount and distribution will modify the radiative structure of the atmosphere over the plateau, thereby influencing the climate, ecosystem and human activities (Forster and Shine, 1997; Hartmann et al., 2000; Fuhrer and Booker, 2003).

Using observations from the Total Ozone Mapping Spectrometer (TOMS) satellite instrument, a persistent summertime total column ozone low (TOL) centred over the TP was reported by Zhou et al. (1995). Later studies using satellite and ozonesonde data also found the ozone low in other seasons but with different magnitudes (Zheng et al., 2004; Bian et al., 2006; Tobo et al., 2008). Zou (1996) analyzed total ozone seasonal variations and trends over the TP and showed that
relative to zonal mean values, the largest ozone deficit occurs in May, while the smallest deficit occurs in wintertime. They also reported a negative correlation between the ozone deficits and the heat flux from the surface to the air over the plateau. Ye and Xu (2003) also confirmed the persistent existence of the TOL over the TP. They proposed that the high topography and the elevated heating source associated with thermally forced circulations are the two main reasons
for its occurrence. In addition, previous observational and modelling studies have suggested that the thermal-dynamical forcing of the TP, for example by air expansion, uplifting of the tropopause, thermal convection, and monsoon circulation, makes a dominant contribution to the TOL especially in summer (Cong et al., 2001; Liu et al., 2003; Ye and Xu, 2003; Tian et al., 2008, 2011; Bian et al., 2011; Guo et al., 2012, 2015; Zhang et al., 2014; Chen et al., 2017).
However, the exact coupling pathways between the thermal-dynamical forcing and long-term total column ozone (TCO) changes during different seasons are still not well established.

It is well known that Antarctic stratospheric ozone decreased severely due to anthropogenic emissions of ozone-depleting substances (ODS) from the 1980s onwards (Farman et al., 1985; WMO, 2003, 2007, 2011 and references therein). Also, following the implementation of the
Montreal Protocol in 1987, signs of an ozone recovery have been reported in recent years (WMO, 2014; Chipperfield et al., 2015, 2017, 2018; Solomon et al., 2016; Kuttippurath and Nair, 2017; Pazmiño et al., 2018; Strahan and Douglass, 2018; Weber et al., 2018). Outside of the polar regions, column ozone amounts are largely determined by the stratospheric dynamics and hence quantifying long-term trends is quite challenging (Newman et al., 1997; Rex et al., 2004;
Manney et al., 2011; Zhang et al., 2016, 2018, 2019). Observations and model simulations indicate that the variability and long-term ozone trends are significantly different at different latitudes (e.g. Austin et al., 2010; Chipperfield et al., 2017, 2018). Major factors contributing to short- and long-term ozone variations include changes in ODS emissions, atmospheric dynamics, solar irradiance and volcanic aerosols (Pawson et al., 2014; Harris et al., 2015).

Previous studies have also documented that TP trends can be affected significantly by internal variabilities. Zou (1996) reported strong negative ozone trends over Tibet for the 1979–1991 time period. The effects of the quasi-biennial oscillation (QBO) and the El Niño-Southern



Oscillation (ENSO) on TCO over Tibet were analyzed in subsequent studies (e.g. Zou et al., 2000, 2001). The stratospheric ozone abundance can also be influenced by long-term variations in volcanic aerosols and solar radiation (Solomon, 1999; Soukharev and Hood, 2006; Fioletov, 2009; Dhomse et al., 2011, 2015, 2016). Besides these traditional explanatory factors, some dynamical proxies, e.g., temperature and geopotential height (GH) have been shown to have significant influence on the long-term ozone variations and effectively help better estimation of ozone trends (e.g. Ziemke et al., 1997; Dhomse et al., 2006). Zhou and Zhang (2005) presented decadal ozone trends over the TP using the merged TOMS/SBUV ozone data over the period 1979-2002 and found that the downward trends are closely related to the long-term changes of temperature and geopotential height. Zhou et al. (2013) found substantial downward ozone trends in the merged TOMS/SBUV ozone data (1979-2010) during the winter-spring seasons over the TP. They also showed that long-term ozone variations are largely affected by the thermal-dynamical proxies such as the lower stratospheric temperatures, with its contribution reaching around 10% of the total ozone change. Zhang et al. (2014) indicated that the TOL over the TP in winter has deepened during the period 1979-2009 and the thermal-dynamical processes associated with the TP warming (increasing surface temperature) account for more than 50% of the TCO decline in this region.

Many previous studies have demonstrated the contributions of dynamical processes to the long-term ozone variation in different latitude bands (Dhomse et al., 2006; Chehade et al., 2014) as well as the TP region (Zhou et al., 2013; Zhang et al., 2014). As we know that wintertime stratospheric circulation has large interannual variability that is mainly driven by tropospheric processes, the choice of dynamical proxy for this tropospheric influence varies for different latitude bands. For mid-high latitudes, most studies use Eliassen-Palm Flux (or heat flux) to explain a large part of dynamical variability (e.g. Weber et al, 2003). However, heat flux is not a suitable dynamical proxy for subtropical latitudes (e.g. Fusco and Salby, 1999; Hood and Soukharev, 2005; Dhomse et al., 2006) as transport in this region is balanced by tropical upwelling and isentropic transport in the lower stratosphere. Hence a better proxy is needed to explain the dynamical influence for the TP region.

Under the background of global surface warming, concern about regional climate changes have been focused on high-elevation areas, such as the Tibetan Plateau. The geopotential height in the free atmosphere is an important thermal-dynamical proxy that not only conveys information about the thermal structure of the atmosphere, but also serves as an indicator of synoptic circulation changes (Lott et al., 2013; Christidis and Stott, 2015). The natural and anthropogenic contributions to the changes in GH establish the coherent thermal-dynamical nature of externally forced changes in the regional climate system, which provides the basis for the validation of climate models. In this study, the GH at 150 hPa over the TP is used as a new thermal-dynamical proxy which incorporates coupling between the local TP circulation and various tropospheric teleconnection patterns and represents the tropospheric dynamical influence more realistically.

With the extended Copernicus climate change service (C3S) TCO time series available from 1979 to 2017, the aim of this paper is to study the long-term ozone trend and variability over the Tibetan region. Based on statistical regression analysis of C3S ozone data and SLIMCAT three-dimensional (3-D) chemical transport model (CTM) simulations, the contributions of different influencing variables including the local thermal-dynamical proxy (GH) are highlighted





to help general understanding of the long-term evolution of the ozone variation in different seasons and over different areas.

The layout of the paper is as follows. Section 2 introduces C3S and SLIMCAT model simulations as well as the regression methods used for the analysis of the total ozone variability.
The long-term TCO and TOL trends over the TP region are presented in section 3. Regression results based on C3S and analysis of the contribution of different explanatory variables for different areas and in different seasons are given in section 4. In section 5, regression and sensitive experiment results based on SLIMCAT are discussed followed by our summary and conclusions in Section 6.

## 2 Data and Methods

### 2.1 C3S

High quality observational based datasets are necessary for better quantification of decadal TCO trends. This is because inter-annual variability can cause variations of up to 20% whereas ozone
trends are generally less than half a percent. As the lifetime of most satellite instruments is less than two decades, merged satellite datasets are widely used to determine long-term ozone trends. These datasets are created by combining total ozone measurements from different individual instruments to provide global coverage over several decades (e.g. Frith et al., 2014). However, such merged satellite data sets are available with coarse resolutions, hence are not well suited to
study relatively small geographical areas such as the TP. Hence, here we use the total column ozone from the Copernicus Climate Change system (C3S) which is implemented by the European Centre for Medium-Range Weather Forecasts (ECMWF). For detailed description and data availability, see https://cds.climate.copernicus.eu/cds. In brief, these are monthly mean gridded data that span from 1970 to present. They are created by combining total ozone data
from 15 satellite sensors including GOME (1995-2011), SCIAMACHY (2002-2012), OMI (2004-present), GOME-2A/B (2007-present), BUV-Nimbus4 (1970-1980), TOMS-Nimbus7 (1979-1994), TOMS-EP (1996-2006), SBUV-9, -11, -14, -16, -17, -18, -19 (1985-present) and OMPS (2012-present). The horizontal resolution of the assimilated product after January 1979 is $0.5° \times 0.5°$. The document describing the methodology adopted for the quality assurance in the
C3S-Ozone procurement service, with detailed information about the ground-based measurements used to verify satellite observations, the specific technical project implemented to compare the gridded (level-3) and assimilated (level-4) data, and the metrics developed to associate validation results with user requirements, can be downloaded from https://cds.climate.copernicus.eu/cdsapp#!/dataset/ozone-monthly-gridded-data-from-1970-to-pr
esent?tab=doc. The strength of this data set is the long-term stability of the total column monthly gridded average product that is below the 1%/decade level. Systematic and random errors in this data are below 2% and 3-4%, respectively, hence making it well suited for long-term trend analysis. The evaluation of ozone trends performed using merged deseasonalized anomalies is presented in Sofieva et al. (2017) and Steinbrecht et al. (2017). They show that ozone trends are
in agreement with those obtained using other datasets, and they are close to those reported in WMO (2014).





Here we use C3S data for the time period 1979-2017. Overall we use four different area-weighted total ozone time series: TP (27.5 °-37.5 °N, 75.5 °-105.5 °E), zonal TP (full zonal mean for 27.5 °-37.5 °N) as well as zonal mean for latitude bands to the south (10 °-20 °N) and north (40 °-50 °N) of the TP region.

## 2.2 TOMCAT/SLIMCAT model

Chemistry-transport models are important tools to investigate how past and present-day ODS and greenhouse gas (GHG) concentrations have influenced the ozone layer (Shepherd et al., 2014; Zvyagintsev et al., 2015). In combination with observed ozone time series, simulations allow the attribution of ozone changes, thus encapsulating our understanding of the fundamental physics and chemistry that controls ozone and its variations (e.g. Chipperfield et al., 2017). TOMCAT/SLIMCAT (hereafter SLIMCAT) is a 3-D off-line chemical transport model (Chipperfield et al., 2006), which uses winds and temperatures from meteorological analyses (usually ECMWF) to specify the atmospheric transport and temperatures and calculates the abundances of chemical species in the troposphere and stratosphere. The model has the option of detailed chemical schemes for various scenarios with different assumptions of factors affecting ozone (e.g. Feng et al., 2011; Grooss et al., 2018), including the concentrations of major ozone-depleting substances, aerosol effects from volcanic eruptions (e.g. Dhomse et al., 2015), and variations in solar forcing (e.g., Dhomse et al., 2016) and surface conditions. For this study, the model has been forced by ECMWF ERA-Interim reanalysis (Dee et al., 2010) and run from 1979-2017 at a resolution of $2.8° \times 2.8°$ with 32 levels (up to around 60 km).

We perform control and sensitivity simulations based on the SLIMCAT CTM to elucidate the impact of dynamical changes on the total ozone variations over the TP region. The control experiment R1 uses standard chemical and dynamical parameters for the time period 1979-2017, which is identical to the control run of Chipperfield et al. (2017). To understand the special dynamical influences (e.g. GH) on ozone variations over the TP, two sensitivity experiments R2 and R3 were performed with all configurations the same as R1 except the simulations used annually repeating meteorology for the years 2004 and 2008, respectively. These years were chosen because the 150 hPa GH in wintertime is substantially different while other dynamical proxies are almost the same for the two years.

## 2.3 Regression methods

To assess the long-term total ozone variations due to various natural and anthropogenic processes, many regression methods have been employed. Models based on Equivalent Effective Stratospheric Chlorine (EESC) or piecewise linear trend (PWLT), as well as other explanatory proxies, are the most widely used (Reinsel et al., 2002; Nair et al., 2013; Chehade et al., 2014). EESC is a measure of the inorganic chlorine and bromine amounts accumulated in the stratosphere (WMO, 2003), which drives chemical ozone depletion. Previous studies have indicated that EESC is a main contributor to the long-term global ozone decline and the trend





changes after the end of 1990s (Newman et al., 2004; Fioletov and Shepherd, 2005; Dhomse et al., 2006; Randel, 2007; Harris et al., 2008; Kiesewetter et al., 2010). We use this method to study the effect of EESC on the long-term ozone variations over the TP and the other zonal regions. We also use a PWLT regression method for comparison with a pair of linear trends to
statistically analyze the decrease and recovery trends in the total ozone over the TP and zonal-TP regions before and after the EESC peak in 1997. Our aim is to clarify statistical significance of the key processes responsible for the total column ozone variations over the TP in different seasons, using two different regression models.

Traditional explanatory variables to account for chemical and dynamical processes in the
atmosphere, include the F10.7 solar flux for the 11-year solar cycle, quasi-biennial oscillation (QBO) at 30 hPa and 10 hPa ($a \times QBO30 + b \times QBO10$), and ENSO (e.g. Baldwin et al. 2001; Camp and Tung, 2007; Xie et al. 2016). Many other studies also include aerosol optical depth (AOD) at 550 nm to account for volcanically enhanced stratospheric aerosol loading as well as Arctic oscillation (AO) to account for high latitude dynamical variability (e.g. WMO, 2016 and
references therein). For the TP region, the local thermal-dynamical forcing, e.g. the geopotential height at 150 hPa (GH150) or the surface temperature (ST) is also considered to better explain the long-term ozone variations (Zhou et al., 2013; Zhang et al., 2014).

**Table 1**. Correlation values for the TCO and explanatory variables over the TP in DJF during 1979-2017

| Corr. | TCO | EESC | Solar | QBO30 | QBO10 | ENSO | Aerosol | AO | GH150 | ST |
|---|---|---|---|---|---|---|---|---|---|---|
| TCO | 1 | -0.299 * | 0.298* | -0.334 ** | -0.543 *** | 0.293 * | 0.066 | -0.121 | -0.542 *** | -0.304 ** |
| EESC | | 1 | -0.195 | 0.029 | 0.039 | -0.064 | -0.138 | 0.119 | -0.058 | -0.057 |
| Solar | | | 1 | 0.011 | 0.060 | 0.035 | 0.234 | 0.396 ** | 0.069 | -0.089 |
| QBO30 | | | | 1 | 0.011 | 0.007 | -0.036 | 0.222 | 0.163 | -0.072 |
| QBO10 | | | | | 1 | -0.011 | 0.219 | 0.100 | 0.097 | -0.069 |
| ENSO | | | | | | 1 | 0.371 ** | -0.181 | -0.468 *** | -0.089 |
| Aerosol | | | | | | | 1 | 0.216 | 0.215 | -0.309 ** |
| AO | | | | | | | | 1 | 0.374 ** | -0.104 |
| GH150 | | | | | | | | | 1 | 0.617 *** |
| ST | | | | | | | | | | 1 |

225               *** 99% confidence level; ** 95% confidence level; * 90% confidence level

Due to the large difference in scales and units of the explanatory variable time series, all the time series are standardized with respect to mean zero and a standard deviation of one. This ensures each factor contributes approximately proportionately to the final ozone variations. The transformation does not change the correlation and fitting coefficients. Before the multiple linear
regression for the long-term ozone variations, we calculated the correlation between total ozone and the explanatory variables. **Table 1** shows the correlation values for the TCO (TP region) and





the explanatory variables averaged for winter (December-January-February, DJF) during 1979-2017. Correlation analysis for the variables during summer months (June-July-August, JJA) is also presented in the supplementary **Table S1**. Both local thermal-dynamical proxies (GH and ST) are de-trended before being used into the regression model. The relationships between the TCO and the explanatory variables are statistically significant except for the aerosol and AO. As shown from the correlation analysis, the DJF mean solar variability is strongly correlated with the AO (0.396) time series. Also, the GH150 time series shows somewhat stronger correlation with the ENSO (-0.468), AO (0.373), and ST (0.617) time series. We also find that aerosol and ENSO are correlated (0.371). Hence, to avoid any aliasing effects and for the better estimation of regression coefficients and their attributing variations in ozone, it is essential to have independent explanatory variables in the regression model. In our regression model, we remove the influence of volcanic aerosols after El Chichón (1982) and Mount Pinatubo (1991) eruptions by leaving out the data in the years of 1982, 1983, 1991 and 1992. AO is also removed as it shows strong correlation with the solar and GH150 proxies. As for the other strongly correlated factors (ENSO, GH150 and ST), we make three groups of independent variables to analyze the Tibetan TCO variations and compare the corresponding regression results under different situations:

$$TCO(t) = C_0 + C_1 \cdot EESC(t) + C_2 \cdot solar(t) + C_3 \cdot QBO(t) + C_4 \cdot ENSO(t) + \varepsilon(t) \qquad (1)$$

$$TCO(t) = C_0 + C_1 \cdot EESC(t) + C_2 \cdot solar(t) + C_3 \cdot QBO(t) + C_4 \cdot ENSO(t) + C_5 \cdot ST(t) + \varepsilon(t) \qquad (2)$$

$$TCO(t) = C_0 + C_1 \cdot EESC(t) + C_2 \cdot solar(t) + C_3 \cdot QBO(t) + C_4 \cdot GH150(t) + \varepsilon(t) \qquad (3)$$

where $t$ is a running index corresponding to the years during the period 1979-2017, excluding the four years due to the volcanic aerosol loading. $C_0$ is a constant for the long-term average. $C_1$-$C_5$ represent the time-dependent regression coefficients of each proxy and $\varepsilon$ is the residual. In the PWLT regression model, the $C_1 \times EESC(t)$ term is replaced by $(c1 \times t1 + c2 \times t2)$ in Eq. (1-3) with linear trends in the periods 1979–1996 and 1997–2017, respectively.

## 3. TCO and TOL trends over the TP

**Figure 1** shows the seasonal mean (DJF and JJA) total ozone time series over the TP (27.5°-37.5°N, 75.5°-105.5°E), zonal-TP (27.5°-37.5°N), South-TP (10°-20°N), and North-TP (40°-50°N) regions. As expected, the South-TP latitude band shows about 16% and 41% less ozone compared to zonal-TP and North-TP zonal bands in DJF, respectively. These differences are much smaller in JJA (-7% and -16%, respectively). Also, nearly all the time series show much larger interannual variations in DJF compared to JJA and variations are largest in the North-TP time series. Another important aspect is that the TP and zonal-TP time series in DJF are close to each other (difference < 5 DU), but shows a significant difference (~20 DU) in JJA. This is consistent with previous studies (e.g. Ye and Xu, 2003; Zhang et al., 2014). Otherwise, compared with JJA time series, total ozone time series over the TP for the other seasons show less differences against the zonal-TP time series, and are discussed in **Figure 2**.





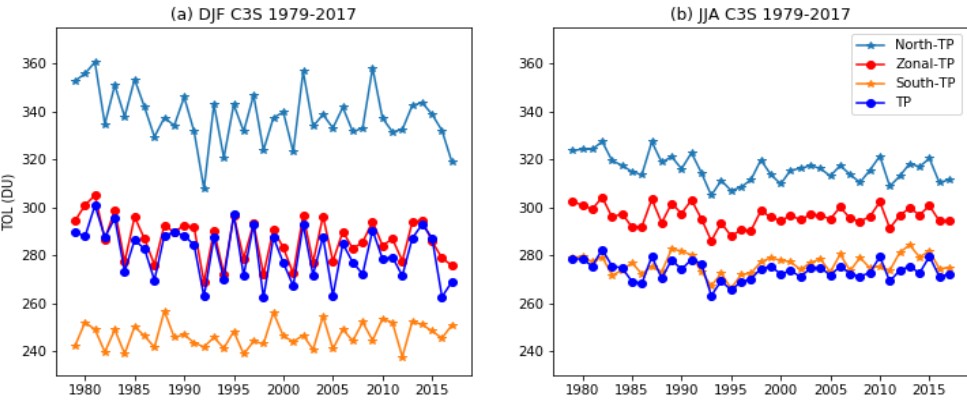


**Figure 1**. Long-term variations of total ozone columns averaged for December-January-February (DJF) and June-July-August (JJA) seasons during 1979-2017 over the TP region (27.5 °-37.5 °N, 75.5 °-105.5 °E, blue line), zonal-TP (27.5 °-37.5 °N, red line) as well as other non-Tibetan regions along with South-TP (10 °-20 °N), and North-TP (40 °-50 °N) zonal bands shown with orange and light blue lines, respectively.


To illustrate TOL characteristics, the zonal deviations of the TCO at each grid point, calculated by subtracting the zonal mean total ozone for each latitude band, are shown in **Figure 2**. These zonal deviations are calculated for the period 1979-2017. The negative zonal deviations suggest that the total ozone low centred over the TP exists for all the seasons. As expected, the TOL over the TP is

most discernible in JJA while weakest in DJF. The TOL centre also moves from the northwest in spring (March-April-May, MAM) to the south in winter (DJF). The dark blue contours over the TP region in summer (JJA) and spring (MAM) show the zonal deficits of more than 20 DU, which are more substantial than those (up to 10 DU) in autumn (September-October-November, SON) and winter (DJF). The TOL over the TP with the difference in wintertime and summertime is mainly

caused by the terrain effect and the dynamic transport effect. In wintertime, the plateau geographic effect probably accounts for the formation of TOL due to the lack of about 4 km air column containing ozone (Ye and Xu, 2003). During summertime, the elevated heating source with rising air over the TP leads to thermally forced anticyclonic circulation (Yanai et al., 1992). The upper-level Asian summer monsoon anticyclone exhibits intraseasonal variability, and its

coupling with deep convection over the TP potentially transports ozone-poor air from the boundary layer upward into the upper troposphere and lower stratosphere (Gettelman et al., 2004; Randel and Park, 2006; Tobo et al., 2008; Bian et al., 2011).



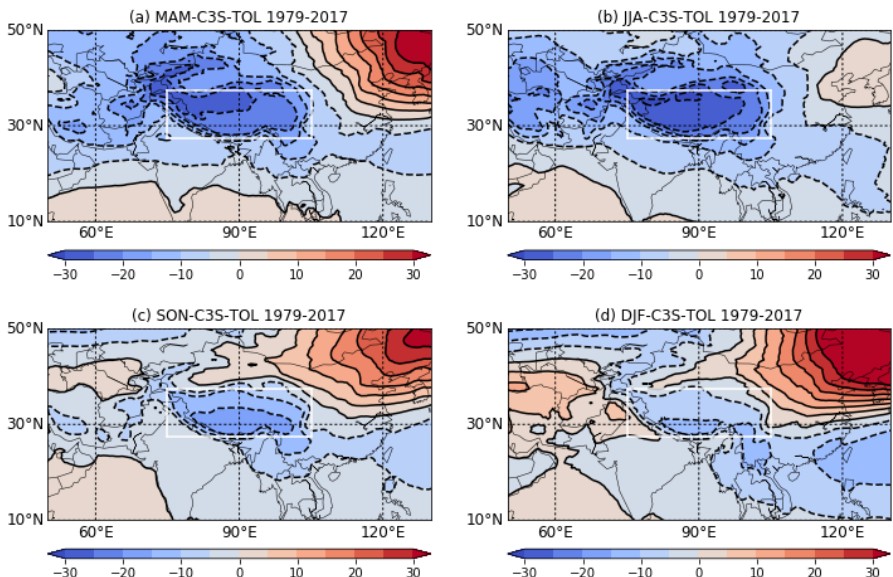

**Figure 2.** Latitude-longitude cross section of the zonal ozone deviations for (a) March-April-May (MAM),
(b) June-July-August (JJA), (c) September-October-November (SON) and December-January-February
(DJF) seasons based on C3S total ozone dataset for 1979-2017 time period. The solid and dashed
contours represent the positive and negative zonal deviations. The contour interval is 5 DU. The TP
region (27.5 °-37.5 °N, 75.5 °-105.5 °E) is marked by the white rectangle.


**Figure 3** shows the monthly TCO climatological values and trends over the TP and the zonal-TP
region for the period 1979-2017. The mean TCO values and the limits of maximum/minimum
range over the TP are smaller than those over the zonal-TP region throughout the year. The
monthly mean TCO over the TP shows a maximum in March (~297 DU) and a minimum in
October (~262 DU), while in the zonal-TP time series maximum and minimum appear in April
(~317 DU) and November (~270 DU), respectively. According to Ye and Xu (2003), this annual
variability is a result of the high topography of the TP causing a weaker amplitude and an earlier
phase (about 1 month). The wintertime ozone buildup and steady summertime ozone decline are
evident over both regions (TP and zonal-TP), which is consistent with the typical total ozone
variations discussed in previous studies (e.g. Randel et al., 2002; Fioletov and Shepherd, 2003).
The TCO trends over the TP and zonal-TP are calculated using an ordinary least square regression
(OLR) model, and are shown in **Figure 3b**. The negative trends of the TCO over the TP are
generally a little stronger than zonal TP region, especially during winter months, consistent with
previous findings (Zhang et al., 2014; Zhou et al., 2014). OLR analysis suggests the largest
negative trend (-0.41 ±0.21 DU/yr) over the TP occurs in February. The weakest trend occurs in
October, when the zonal-TP time series show a relatively more obvious decline (-0.10 ±0.05
DU/yr). This is somewhat consistent with the finding by Fioletov and Shepherd (2003) who





showed that the long-term ozone trends over northern midlatitudes are in line with the interannual variability and are largely determined by the negative trends in the winter-spring ozone buildup.


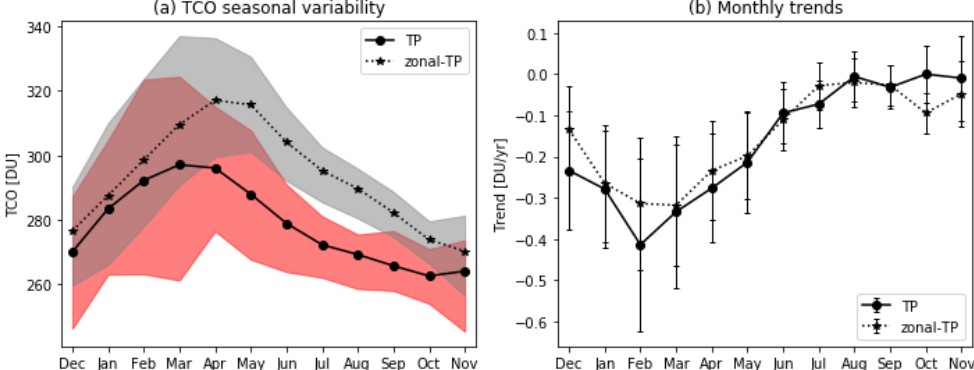

**Figure 3**. (a) Seasonal variations in TCO during 1979-2017 over the TP (solid circles, 27.5-37.5°N, 75.5-105.5°E) and the zonal-TP region (asterisks, 27.5-37.5°N). The red and grey shaded areas show the maximum-minimum TCO ranges for the TP and the zonal-TP region; (b) Linear trends calculated using ordinary least square regression (OLR) method over the TP (black solid line) and the zonal-TP region (black dashed line).

Following monthly TCO trends using OLR, we assess seasonal trends using OLR for both the TP and zonal-TP region. The trend deviations for the DJF mean TCO between the TP and zonal-TP reflect that the TP time series show a stronger negative trend (-0.29 ±0.13 DU/yr) compared to the zonal-TP time series (-0.22 ±0.13 DU/yr), indicating a trend for a deepening in TOL.

**Figure 4** shows the TOL trends for different seasons with 95% confidence bounds and 95% prediction bounds. Springtime (MAM) and summertime (JJA) trends are weaker in magnitude and statistically insignificant within 2σ. Also, in winter (DJF) the TOL trends are much stronger (-0.67 ± 0.57 DU/decade), although they are still insignificant within 2σ. On the other hand, the TOL time series in autumn (SON) shows positive trends that are statistically significant (0.42 ±0.26 DU/decade). This somewhat conflicting nature of TOL trends as well as the different TOL magnitudes in different seasons could be explained by the fact that winter time ozone concentrations are largely controlled by dynamical processes (ozone build-up), while photochemical loss dominates in summer. Thus, it is necessary to analyze the influences of the chemical and dynamical processes (e.g. EESC, solar, QBO and the local thermal-dynamical proxy) on the total ozone variability under different atmospheric conditions.



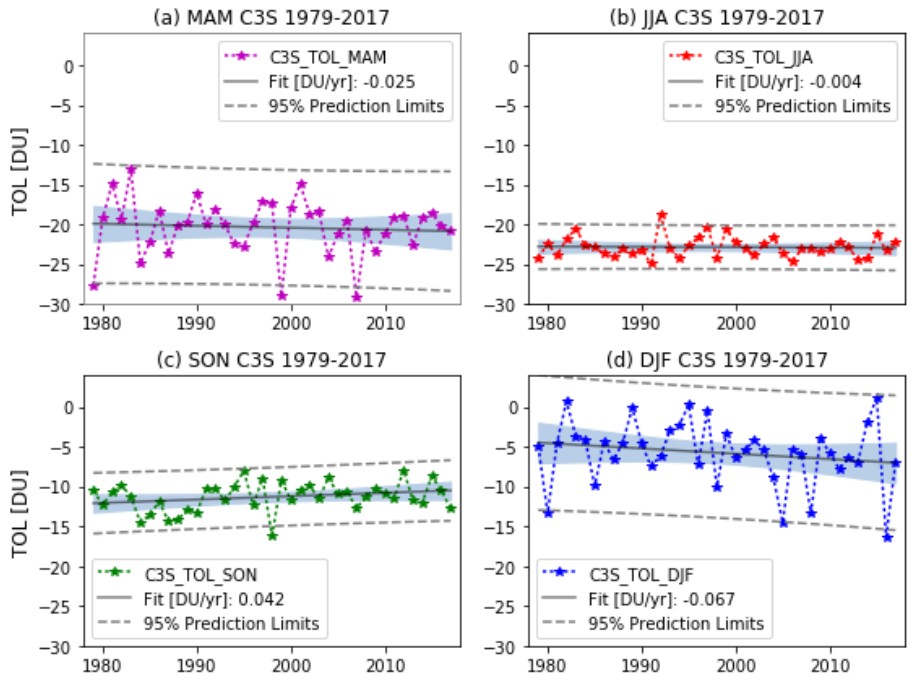

**Figure 4**. Linear regression of the trends of TOL over the TP in different seasons (MAM, JJA, SON and DJF) based on C3S during 1979-2017. The confidence band in the shaded area indicates the trend uncertainty, while the prediction band between the dashed lines is the region that contains approximately 95% chance of a new measurement falling within the band.

## 4 C3S regression results

We apply the EESC-based and PWLT regression models to the C3S TCO time series to quantify the processes controlling interannual ozone variations and the long-term ozone trends over the TP, zonal-TP, South-TP and North-TP zonal bands. Determination coefficients (R-squares) based on EESC-based and PWLT regression models for DJF mean TCO time series over the TP, Zonal-TP, South-TP and North-TP regions are given in **Table 2**. These coefficients are derived using the EESC-based and PWLT methods with three groups of explanatory factors.

Generally, the PWLT regression results are consistent with EESC-based regression results in different situations (e.g. regions or factors), but have better determination coefficients. The comparison of the regression results based on Eq. (1) and Eq. (2) indicates that the additional consideration of the surface temperature (ST) into the model improves the determination coefficients over the different zonal regions, especially over the TP. Consistent with Zhang et al. (2014), surface temperature over the TP improves the regression fit of the long-term ozone





variations, but our analysis suggests that geopotential height at 150 hPa (GH150) is a better dynamical proxy for the TP region.

**Table 2.** Determination coefficients derived using two different regression models (EESC-based and PWLT (in brackets)) when applied to DJF mean TCO time series over different regions

| DJF TCO (R-square) | EESC (Linear trends), solar, QBO, **ENSO** based on Eq. (1) | EESC (Linear trends), solar, QBO, **ENSO, ST** based on Eq. (2) | EESC (Linear trends), solar, QBO, **GH150** based on Eq. (3) |
|---|---|---|---|
| TP, 27.5 °-37.5 °N, 75.5 °-105.5 °E | 0.601 (0.626) | 0.725 (0.761) | **0.780 (0.801)** |
| North-TP, 40 °-50 °N | 0.526 (0.589) | 0.603 (0.657) | 0.544 (0.585) |
| Zonal-TP, 27.5 °-37.5 °N | 0.660 (0.662) | 0.737 (0.760) | **0.747 (0.755)** |
| South-TP, 10 °-20 °N | 0.640 (0.714) | 0.712 (0.771) | 0.656 (0.739) |

The regression analysis based on Eq. (2) and Eq. (3) indicates that inclusion of GH at 150 hPa as a dynamical proxy substantially improves R-square values for both the TP and zonal-TP regions,
but surface temperature seems to perform better for South-TP and North-TP latitudes. This might suggest that the changes of GH at 150 hPa are relevant locally for the ozone variations that are associated with the special orography and local circulation over the TP region. This thermodynamic effect related to the changes in the temperature of the layer between the surface and a given pressure level (150 hPa) is thought to be a more prominent phenomenon (e.g.
Christidis and Stott, 2015).

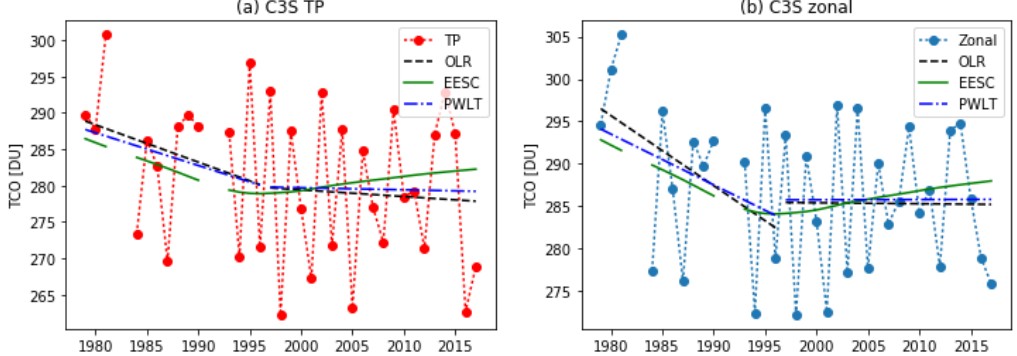

**Figure 5.** Long-term TCO time series (in DU) and the determined trends over (a) the TP and (b) zonal-TP region in DJF during the time periods of 1979-1996 (with 4 years of data removed) and 1997-2017 based on
OLR, EESC-based and PWLT regression (Eq. (3)) models.





**Table 3.** Linear trends in the periods 1979-1996 and 1997-2017 based on OLR, EESC-based and PWLT regression models

| [DU/yr] | OLR | | EESC | | PWLT | |
|---|---|---|---|---|---|---|
| Periods | 1979-1996 | 1997-2017 | 1979-1996 | 1997-2017 | 1979-1996 | 1997-2017 |
| TP | -0.51 ±0.49 | -0.09 ±0.39 | -0.47 ±0.25 | 0.18 ±0.10 | -0.45 ±0.28 | -0.02 ±0.19 |
| Zonal | -0.82 ±0.45 | -0.01 ±0.29 | -0.54 ±0.22 | 0.20 ±0.08 | -0.60 ±0.27 | 0.00 ±0.18 |

**Figure 5** shows the long-term DJF mean TCO trends over the TP and the zonal-TP region for two different time periods: 1979-1996 and 1997-2017 based on three regression methods. OLR and PWLT based linear trends for 1979-1996 and 1997-2017 are listed in **Table 3**. We also present EESC-related ozone trends approximated as the slopes of the EESC trends multiplied by the regression coefficients. All three methods show a stronger negative trend for the zonal-TP region

than the TP for the period 1979-1996 and the trends over both regions are statistically significant. For recent period 1997-2017, the trends based on OLR are still decreasing with a stronger rate of –0.09 ±0.39 DU/yr over the TP and a weaker rate of –0.01 ±0.29 DU/yr over the zonal-TP region. In contrast to the significant EESC-related ozone recovery since 1997, the PWLT regression model still shows a persistent (but statistically insignificant) negative trend over the TP (-0.02 ±

0.19 DU/yr), which is similar to, but slightly weaker than, the OLR model. For zonal-TP TCO time series, PWLT does not show any trend after 1997, whereas the OLR model shows a slightly negative (but insignificant) trend. The determination coefficient of the PWLT regression based on Eq. (3) (0.801) is better than that in the EESC-based regression model (0.780), which indicates that the linear trends based on PWLT method are probably more reasonable.


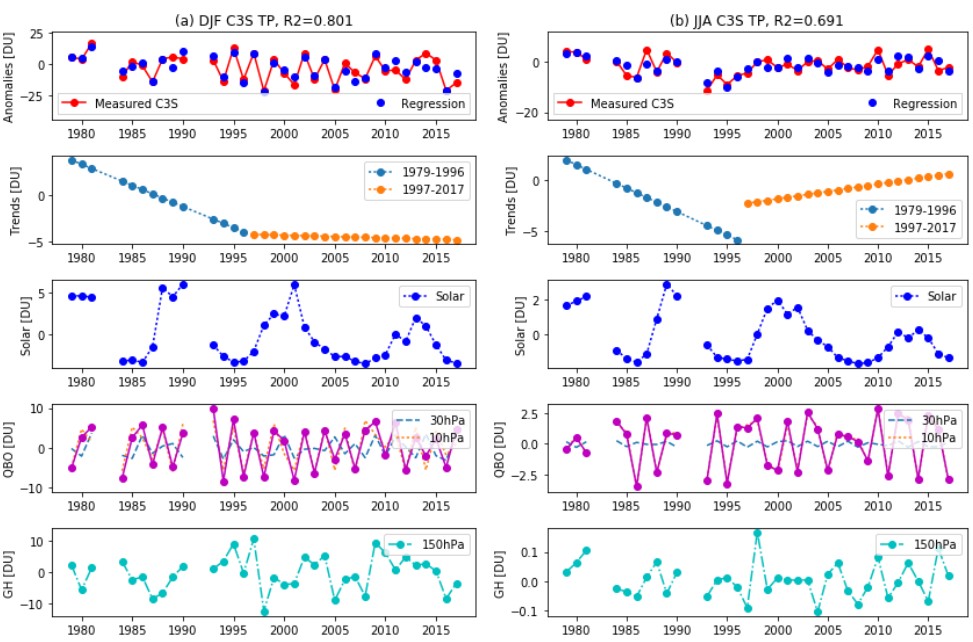





**Figure 6.** (a) PWLT regression results with contributions from linear trends for 1979-1996 and 1997-2017 time periods, solar cycle, QBO, and the 150 hPa GH in DJF based on C3S during 1979-2017 over the TP region; (b) Similar to (a) but with all factors averaged in JJA.

**Figure 6** shows TCO time series for the TP region and PWLT based linear regression fit for both DJF and JJA seasons. Overall, the regressed ozone time series in DJF shows a better fit with the measurement values than that in JJA. The residual mean square (RMS) is 4.60 DU in DJF and 2.13 DU in JJA, which are associated with the seasonal cycle amplitudes for the two different seasons. As the ozone variability is less in summer and autumn than during the seasonal ozone buildup

period in winter and spring (WMO, 2007), the long-term ozone anomalies are smaller in JJA with smaller contributions from different explanatory variables than those in DJF. To quantitatively describe the contributions of those different explanatory variables to the long-term TCO anomalies, the fitted signals of each explanatory term in Eq. (3) are also presented in **Figure 6**. The corresponding regression coefficients with standard deviation are listed in **Table 4**. EESC-based

regression coefficients for the TP region in both DJF and JJA are presented in the supplementary **Table S2**. The linear trend during 1979-1996 over the TP shows an almost similar decline in JJA to that in DJF but a stronger recovery signal after 1997. More importantly, in DJF the 11-year solar cycle contributes up to 8 DU total ozone variability from solar minimum to solar maximum but that is almost half in JJA (~4 DU). The combined QBO at 30 hPa and 10 hPa, fluctuating from

easterly to westerly phases, makes a large contribution to the interannual or even biannual ozone variability. In DJF, both QBO at 30 hPa and 10 hPa make significant contributions to the ozone variations, but in JJA the QBO at 10 hPa dominates significantly. The de-trended 150 hPa GH over the TP makes the highest contribution to the wintertime ozone variability but in JJA the contribution is small and insignificant.


**Table 4.** Regression coefficients of the proxies with standard deviation based on PWLT

| PWLT | DJF ($R^2$=0.801) | | JJA ($R^2$=0.691) | |
|---|---|---|---|---|
| | coef | std err | coef | std err |
| Linear1 | -0.45 | 0.28 | -0.46 | 0.13 |
| Linear2 | -0.02 | 0.19 | 0.15 | 0.09 |
| Solar | 3.17 | 1.01 | 1.40 | 0.49 |
| QBO30 | -2.34 | 0.95 | -0.20 | 0.40 |
| QBO10 | -4.85 | 0.90 | 2.11 | 0.44 |
| GH150 | -5.28 | 0.91 | 0.06 | 0.46 |





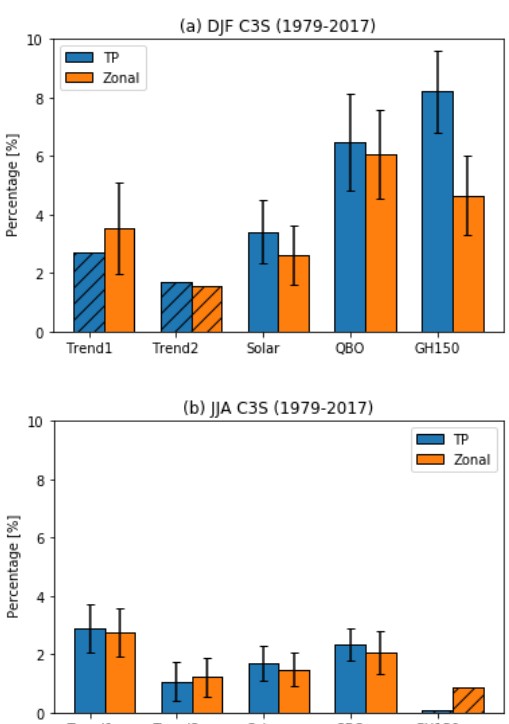

**Figure 7**. Contributions of various explanatory variables to the total ozone variability (in %) in (a) DJF and
(b) JJA over the TP and the zonal-TP region based on C3S data during 1979-2017. The hatched bars
indicate the contribution is not significant within 2σ level.

The contributions of different explanatory variables to the total ozone variability over the TP as
well as the zonal-TP region in DJF and JJA are presented in **Figure 7**. These contributions are
represented by the percentage ozone change as in Eq. (4):

$$\Delta O_3[\%] = \frac{\max(X[DU]) - \min(X[DU])}{mean(TCO[DU])} \times 100\%$$

(4)

where X means the contribution of one proxy (in DU) to the long-term ozone variability. As shown
in **Figure 7**a, dynamical factors (QBO and GH150) exert a relatively stronger effect on the DJF
mean total ozone variability both over the TP and zonal-TP region. The GH at 150 hPa contributes
up to 8% to the total ozone variability over the TP which is even more than that from QBO
(6%).Over the zonal-TP region, QBO dominates with a 6% contribution. In JJA, however, the
contribution from the 150 hPa GH is much smaller than those from the other proxies, especially
over the TP region (0.1%).





Previous studies have found that a strengthened GH associated with an enhanced South Asian high
(SAH) would result in TCO deviations at 150-50 hPa over the TP (Tian et al., 2008; Bian et al.,
2011; Guo et al. 2012). As the SAH approaches and stays over the TP in mid-April, the 150 hPa
GH is much higher in summertime than in wintertime (**Figure S1**). The 150 hPa GH difference
between the TP and zonal-TP region shows a maximum in May when the negative TOL is also
strongest, with a correlation coefficient of -0.86 within the 0.001 significance level. Thus, SAH
imposes an important impact on the formation of the summertime TOL over the TP. However, the
present study shows that the 150 hPa GH makes a major contribution to the wintertime TCO
variability instead of the summertime one. The sharp contrast between the contributions of the 150
hPa GH in DJF and JJA is an interesting feature and possible explanation for those differences is
discussed below.

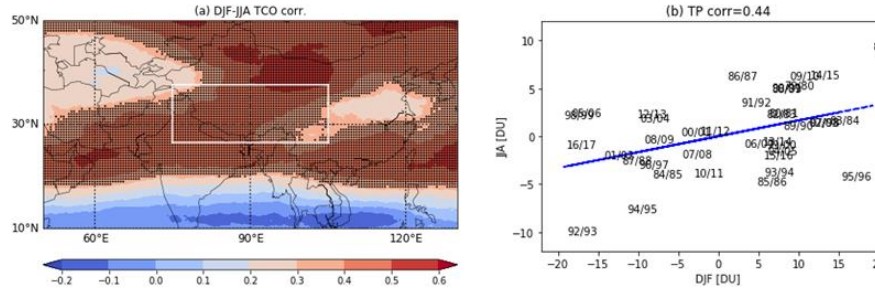

**Figure 8.** (a) Correlation map of the DJF and JJA mean TCO based on C3S during 1979-2017. Correlation
values in the stippled area are statistically significant above 95% confidence level. The white rectangle
represents the TP region. (b) Correlation fit between the DJF and JJA mean ozone anomalies during
1979-2017 over the TP region.

As shown in **Figure 3**, the seasonal variability of the TCO over the TP indicates a marked seasonal
cycle with a buildup of total ozone through the winter and a decline through the summer. Fioletov
and Shepherd (2003) studied the seasonal persistence of midlatitude total ozone anomalies and
demonstrated that ozone values are correlated through the annual cycle from the buildup in
winter-spring to the ozone minimum in autumn. **Figure 8a** shows the correlation of the TCO in
DJF with the subsequent JJA over the northern hemisphere including the TP region during
1979-2017. Correlations that exceed 0.316 are statistically significant above the 95% confidence
level. The significant positive correlation values over the TP region indicate that negative or
positive anomalies seen in wintertime appear to persist through the summer period. As shown in
**Figure 8b,** correlation of the area-weighted total ozone anomalies over the TP in DJF with those in
JJA is 0.44 and is statistically significant. **Table 5** shows the correlation coefficients between
ozone values in a given season of the year with ozone values in subsequent seasons. The decreased
correlation from the buildup in winter to the end of summer also indicates the predictive capability
of ozone concentrations throughout the year. The sharp drop between the summer (JJA) and
autumn (SON) reflects that dynamical variability is nearly absent during summer months and
ozone simply drops off photochemically in a predictable way (Fioletov and Shepherd, 2003).





More detailed correlation of the ozone values between subsequent months of the year has been provided in the supplementary schedules (**Table S3**).


**Table 5**. Correlation coefficient between ozone values in a given season and the subsequent season (1 lag=3 months, bolded numbers are statistically significant within 2σ)

|  | 1 | 2 | 3 |
|---|---|---|---|
| SON | **0.626** | **0.537** | **0.345** |
| DJF | **0.812** | **0.440** | -0.217 |
| MAM | **0.662** | 0.053 | -0.158 |
| JJA | **0.413** | 0.018 | 0.058 |

The seasonal persistence of ozone anomalies over the TP implies a causal link between the
wintertime ozone buildup due to planetary-wave induced transport and the subsequent chemical loss. Previous studies have indicated the ozone buildup in wintertime when transport dominates is modulated by the tropical zonal winds as manifested in the QBO (Holtan and Tan, 1980). In our study, there also exists a clear QBO influence on the wintertime ozone variability over the TP and the zonal region, and moreover a more significant influence over the TP comes from the 150 hPa
GH. In JJA, however, dynamical impact decays and photochemical processes become more important.

## 5 SLIMCAT simulation results

We also apply the PWLT regression analysis in Eq. (3) to the SLIMCAT modelled TCO dataset
during 1979-2017 obtained from the control experiment R1. **Figure 9** shows the contributions (in %) of different proxies to the ozone variability over the TP and the zonal-TP region in DJF and JJA. In DJF, dynamical factors (QBO and GH150) still make relatively large and statistically significant contributions over the TP region. However, in contrast to the results based on C3S in **Figure 5**, the contribution of the 150 hPa GH is much smaller than that of QBO both over the TP
and the zonal region. This difference is probably due to the fact that model simulations are performed at a much coarser resolution ($2.8^{o} \times 2.8^{o}$, 32 levels), which may not be able to represent small scale features such as stratosphere-troposphere exchange as well as tropopause folds realistically. Another important aspect is the inhomogeneities in ERA-interim data, especially before 2000 (e.g. Dhomse et al., 2011, 2013; McLandress et al., 2014). When compared with the
contributions in DJF, the contribution of the 150 hPa GH in JJA also drops sharply over the TP and the zonal-TP region and both are statistically insignificant. The QBO at 30 hPa and 10 hPa still make significant contributions to the ozone variability over the TP and the zonal region, which is probably associated with the transport that dominates in wintertime ozone buildup and persists through the summer period.






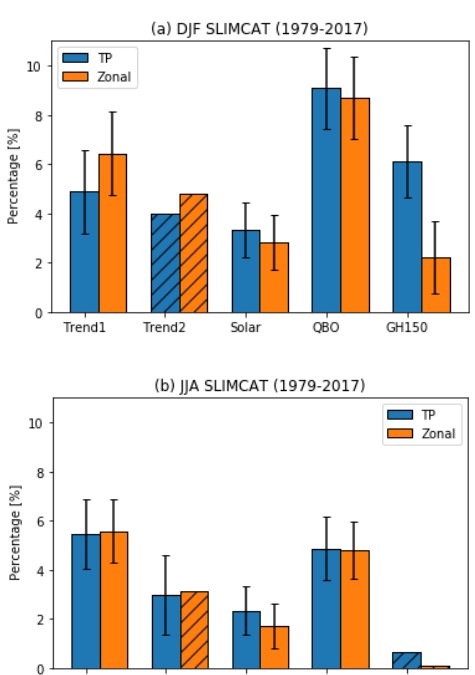

**Figure 9**. Similar to Figure 5 but based on SLIMCAT modelled TCO dataset during 1979-2017.

To further elucidate the role GH at 150 hPa plays in the total ozone variability over the TP, we perform two sensitivity experiments with repeating dynamics from years 2004 and 2008 (R2 and R3), respectively. As the ozone lifetime over the midlatitudes is longer than a few years, we take a 5-year average based on a time slice simulation for 2004-2008 to investigate the ozone variations under different dynamical conditions. Because the mean period of QBO is about 28-29 months, the

difference of QBO between the years of 2004 and 2008 is found to be very small. We propose the GH is a major dynamical contributor to the ozone changes in the two sensitivity experiments. A caveat is that none of the dynamical processes are independent. The GH proxy represents tropospheric dynamical influence somewhat realistically as it incorporates coupling between various tropospheric teleconnection patterns and the local TP circulation.



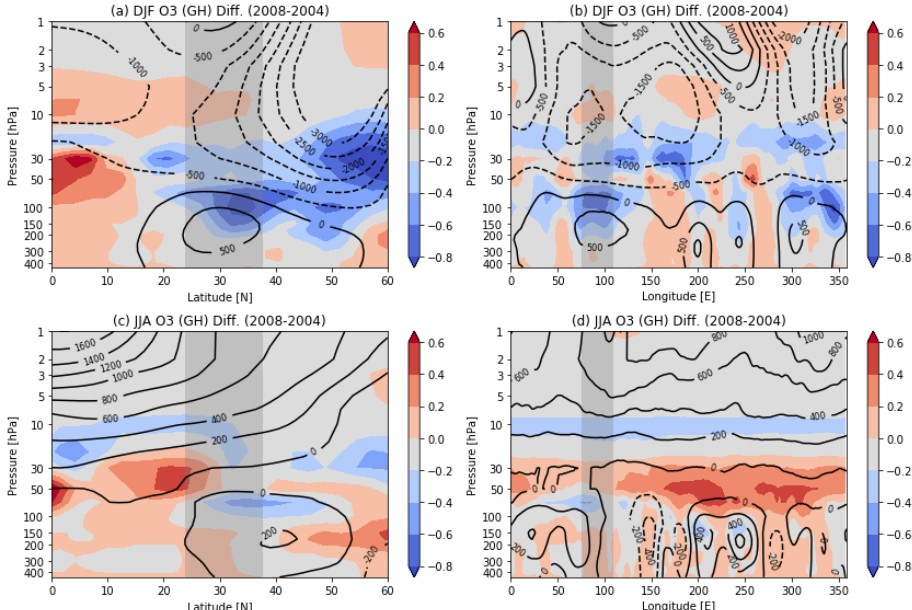


**Figure 10**. (a-b) Pressure-latitude (longitude) cross section of 5-year averaged ozone differences (red and blue colours in DU) in DJF based on SLIMCAT simulations (2.8 ° latitude × 2.8 ° longitude). Solid and dashed lines indicate geopotential height differences (1.5 ° latitude × 1.5 ° longitude). (c-d) Similar to (a-b) but averaged in JJA. Positive ozone and GH differences are shown with red colours and solid lines, whereas blue colours and dashed lines indicate negative differences. The shaded area shows the TP region.


To better understand the zonal and meridional pathways, the vertical GH differences between the two years (2008-2004) along the same longitude band (75.5°-105.5°E) or latitude band (27.5°-37.5°N) as the TP region are presented by the contours in **Figure 10**. The shaded area shows

the TP region over which a most obviously positive anomaly centre of the DJF mean GH differences occurs near the 150 hPa pressure level, as shown in **Figure 10** (a) and (b). The 5-year averaged ozone differences based on the SLIMCAT simulations with fixed dynamics in the two years are also represented by the colours in **Figure 10**. In DJF, there also exists a negative ozone anomaly centered over the TP, which is close to the positive GH anomaly centred at 150 hPa.

While in JJA, neither the GH nor the averaged ozone profiles show a distinct anomaly centre over the TP, as shown by **Figure 10** (c) and (d). By comparing the GH differences along the latitudes, we find that the DJF mean GH differences over the TP are mainly influenced by those over the high latitudes, and in JJA they are mainly influenced by those from low latitudes. This may be because that the TP lies near the boundary between the tropics and midlatitudes in the troposphere.

Due to the fluctuation of the Inter Tropical Convergence Zone (ITCZ), the TP in wintertime is located in midlatitude band where ozone variability is determined by the tropopause height or folds in the lower stratosphere, while in summer, the TP lies in the tropical band where ozone





variability is largely determined by QBO (and QBO-induced circulation) in the mid-stratosphere (Baldwin et al., 2001).

GH differences along the longitudes suggest a tropospheric coupling between the local TP circulation and some tropospheric teleconnection patterns (e.g. ENSO or Walker circulation). As the TP is an elevated heat source, the differences in heat distribution between the plateau and ocean will cause air motions in the zonal and vertical direction. In the normal condition, the pressure gradient force that results from a high-pressure system over the eastern Pacific Ocean
and a low-pressure system over the TP will cause the global general circulation (such as Walker circulation) and therefore affect the ozone distribution. Correlation analysis in Sect. 2 shows that the 150 hPa GH over the TP is in a strong, negative relation to ENSO in DJF, which means during an El Niño event, GH near the TP also increases, thereby increasing tropopause height, leading to a decrease in TCO over the TP. Longitudinal cross section differences show
positive-negative vertical band-like features which seem to closely resemble Walker circulation type anomalies. They also explain why the ozone differences over the TP and the Pacific Ocean are opposite. Thus, GH fluctuations associated with ENSO events or Walker circulation play a key role in controlling TCO variability by altering tropopause height or folds in the lower stratosphere (Piotrowicz et al., 1991; Hu et al., 2106).


## 6 Summary and Conclusions

We have analyzed the variations and trends of the TCO and TOL over the TP in different seasons using the most recent C3S total column ozone data during 1979–2017. We applied two regression methods (PWLT and EESC-based) to analyze the contributions and trends associated with the
dynamical and chemical processes that modify the total ozone changes over the TP and zonal areas. In contrast to conventional regression models, in this study we use a local thermal-dynamical proxy (GH) as a proxy to account for dynamical influence on the wintertime Tibetan ozone changes. We have also performed SLIMCAT 3-D model simulations to explore the role 150 hPa GH plays in the Tibetan ozone variations in different seasons.

Our main conclusions are as follows:

(1) The extended C3S ozone data up to the end of 2017 consolidates the downward linear trends of TCO over the TP region, especially in winter (DJF) with a negative trend of $-0.29 \pm 0.13$ DU/yr. The TOL between the TP and the same latitude band exists throughout a year with the strongest deficit in summer (more than 20 DU) and the weakest deficit in winter (less than 10 DU). The
negative TOL trend in winter (DJF) is deepening more obviously than in other seasons with a rate of $-0.67 \pm 0.57$ DU/decade.

(2) We apply three groups of independent climate variables to the multiple linear regression (MLR) models: EESC-based and PWLT model based on the C3S data during 1979-2017. To avoid the strong correlation between the explanatory variables, the aerosol influence in the years of 1982,
1983, 1991 and 1992 as well as AO is removed. The PWLT model with explanatory factors including EESC, solar cycle, QBO at 30 hPa and 10 hPa and the GH at 150 hPa shows better determination coefficients in DJF for the TP (0.801) and the zonal-TP region (0.755).





(3) The linear trends in the periods 1979-1996 and 1997-2017 from the OLR, EESC-based and PWLT methods show a stronger decline (recovery) signal over the zonal region than that over the TP during 1979-1996 (1997-2017). For the period 1997-2017, the TCO trend based on PWLT shows a slightly decreasing signal (-0.02 ±0.19 DU/yr) over the TP and no recovery signal over the zonal-TP region, which is different from the EESC-based recovery trends and close to the decreasing trends based on OLR.

(4) Based on PWLT regression, dynamical factors (GH150 and QBO) make the major contribution (8% and 6%) to the total ozone variability over the TP. In JJA, QBO still dominates but the 150 hPa GH only contributes 0.1%. The correlation between DJF and subsequent JJA (0.44) indicates the seasonal persistence of total ozone anomalies through the annual cycle from the ozone buildup in winter to the decreasing period in summer. In other words, the dynamical processes (GH150 and QBO) dominate the ozone buildup in wintertime and influence the wintertime ozone variability over the TP. In JJA, the role of dynamical processes becomes insignificant but photochemical loss dominates.

(5) Based on the SLIMCAT regression, we also find that the contribution from the GH at 150 hPa to the ozone variations is more significant in DJF than in JJA. In contrast to C3S regression results, the QBO in both seasons makes a dominant contribution to the total ozone variability. The GH differences between the two years used for sensitivity experiments (2008-2004) show an obvious, negative centre near 150 hPa over the TP in DJF. The 5-year averaged ozone differences based on the two sensitive experiments also show an obvious, positive centre over the TP. Composite analysis show that GH fluctuations associated with ITCZ, ENSO events or Walker circulation play a key role in controlling TCO variability in the lower stratosphere.

Overall, our results show that despite the onset of global stratospheric ozone recovery, column ozone over the Tibetan Plateau is continuing to decline. The implication of this for the local climate needs further investigation.

*Data availability.* The satellite and climate data used in this study are available at the sources and references in the dataset section. The model data used are available upon request (w.feng@ncas.ac.uk).

*Author contributions.* YL performed the data analysis and prepared the manuscript. MPC, WF, SSD, RJP, GD and FL gave support for discussion, simulation and interpretation, and helped improve the paper. All authors edited and contributed to subsequent drafts of the manuscript.

*Competing interests.* The authors declare that they have no conflict of interest.

*Acknowledgements.* We are grateful to the Copernicus Climate Change Service (C3S) for providing the global ozone dataset. TOMCAT/SLIMCAT is supported by National Centre for Atmospheric Science (NCAS). We thank all providers of the climate data used in this study. We thank Jiankai Zhang (Univ. Cambridge) and Dingzhu Hu (Univ. Reading) for helpful suggestions on the Tibetan ozone trends and regression analysis. We also acknowledge the support of



National Natural Science Foundation of China (Grant No. 41127901), Jiangsu provincial government scholarship programme and the Natural Science Foundation for universities in Jiangsu province (Grant No. 17KJD170004).

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
