# Peer review of "Decreases in wintertime total column ozone over the Tibetan Plateau during 1979-2017"

_Atmospheric Chemistry and Physics, 2019_

## Referee Comment (RC1) · Anonymous Referee #1 · 28 Nov 2019

**1   Overall Remarks**

The manuscript investigates interannual, as well as long-term variations of total ozone columns over the Tibetan Plateau and neigbouring regions. Analysis is based on the Copernicus Climate Change Service data-set, and on SLIMCAT chemical transport model simulations. Direct observational data are not used. Main analysis tool is multiple linear regression. In principle this could provide some new and useful information to ACP readers. However, I found it very difficult to grasp meaningful main messages from the manuscript. In my opininion the manuscript presents a largely un-organized smorgas-board of regression results, which may or may not be statistically significant. Results are taken largely at face-value. I did not see any clear scientific questions to

be addressed, nor any stringent logical arguments towards answering such questions. In a similar fashion, the manuscript does cite a large number of papers, but I never see any coherent line-of-thought, how the present results would add new knowledge to what is already out there.

In my opinion the manuscript needs a major rewrite and re-organization. The authors should first decide on their new and major results and then work out these main messages from their analysis. A much more clear and concise presentation is necessary. Right now, I feel that the most appropriate title for the manuscript would be "Regressions that we did, and correlations that we found, for total ozone data near the Tibetan Plateau". This might be OK for a bachelor-thesis, but certaintly not for an ACP paper.

**2  Detailed Suggestions**

Abstract: EESC says TP ozone should go up since 1997, PWLT and OLR say ozone is going down or staying constant. Does this difference mean anything? Is it even statistically significant? Is EESC even a good / relevant proxy for TP ozone? Does the GH150 proxy explain some of the ozone trend? Is there a trend in the GH150 proxy (or in surface temperature)? Is that relevant for the ozone trend? Do the SLIMCAT experiments provide more information about the underlying processes than the C3S data? The fact that regression of SLIMCAT data gives very similar results to regression of C3S data does not provide new insight. Both should give very similar results, because underlying metorological conditions are very similar, and SLIMCAT should represent chemistry reasonably well. Does the comparison between 2004 and 2008 in SLIMCAT data provide anything different from comparing 2004 and 2008 in C3S data? All these questions come up when reading abstract and paper. None of them is answered satisfactorily.

Section 2.1: C3S is based on model assimilation of meteorological and ozone observations. It would be very important to check if the results in the paper are also valid for real ozone observations (e.g. from the SBUV or GOME series of satellites, or, even better, from a nearby total ozone station) and for real geopotential heights, e.g. from a nearby radiosonde station. Such data should be added to the manuscript / plots.

Section 2.2: Are the SLIMCAT model results really important for the main messages of the paper? If not, maybe just omit them. To me, it never becomes clear what additional insight comes from the (coarser resolution than C3S) SLIMCAT simulations here.

Section 2.3: I found this section, especially the results part, out-of-place and confusing here. To me, a more logical flow would be to present the total ozone time-series first, then, later, the multiple linear regression and its results.

Figure 3 (especially 3b) and Figure 4: How relevant are these for the main new messages of the paper? Is there much value in a simple linear trend over the entire period? Especially, using and discussing this simple linear trend, to me, creates confusion and mis-understanding for the later use of the more comprehensive regression (which includes the additional proxies). I would suggest to omit both Figures and their corresponding text.

Beginning of Section 4: That would be the right place for the description of the MLR / Equation 2 and for the presentation of Table 1.

Figure 10: How would that Figure look for C3S data? How would it look for a single model year? I would assume that each SLIMCAT year would be very similar for repeating meteorology. Discussion of Fig. 10. From Fig. 10 it is not clear to me what causes what. Especially in JJA, there are really no distinctive features near the TP. If there are signatures of circulation cells, e.g. with locally high / low ozone, these circulation cells (Walker, ENSO) should be indicated (e.g. by arrows) in the Figure. As it is now, the discussion of Fig. 10 is largely speculative / hand-waving. Not very convincing to me.

Table 1, 2. Are these correlations and regressions taken for the individual months D, J,

F, or for the 3-month DJF average. From the Figures it seems like the latter is the case. However, this should clarified in the Table captions.

Table 2 and its discussion: Increasing $R$ or $R^2$ with the addition of new predictors is quite normal. Every additional predictor is likely to pick up some variance, but this may be random and meaningless. This might be the case for the slightly larger $R$ for PWLT over EESC, because PWLT or 2-Linear-Trends add additional predictors. So the authors should be careful not to overinterpret small changes in $R$. Try it, take any additional random time series, and add it as a predictor. It will increase the overall $R$. So the real question is: Is the increase in $R$ significant? There are tests for this (e.g. F-test, or adjusted $R$). These should be used, and small changes in $R$ should not be over-interpreted.

Table 3: Are the stated uncertainties $1\sigma$ or $2\sigma$? Please state. Even with $2\sigma$, many of the discussed difference would, at the most, be borderline significant. With $1\sigma$, uncertainties should be doubled, and nearly all statistical significance would disappear.

Table 4: Again, is the given uncertainty / std. err $1\sigma$ or $2\sigma$?

Fig. 8 and its discussion: Does this add anything substantial over what is already known, e.g. from *Fioletov and Sheppard* (2003) or *Holten and Tan* (1980)? If not, does it add anything to the salient main messages of the paper? If not, maybe just drop it?

Overall, I think this manuscript needs substantial revision to meet the standards for publication in ACP. Right now, it is too much of a smorgas-board thesis extract.

---

## Short Comment (SC1) · 7 Jan 2020

1.ÂăÂăÂăÂă Is there any results or discussion of TOL in the abstract? 2.ÂăÂăÂăÂă Please explain the reason why choosing 4 TP regions? 3.ÂăÂăÂăÂă Does Fig.1 show results of C3S? 4.ÂăÂăÂăÂă In Fig.6, for QBO analysis, please make sure whether purple dots represent combined QBO at 30hPa and 10hPa? 5.ÂăÂăÂăÂă SLIMCAT results show much smaller 150hPa GH contribution in DJF due to coarser resolutions? Simulations with a finer resolution might be suggested to perform here. The values in JJA almost double in model simulations. It might need some discussions.

---

## Referee Comment (RC2) · Anonymous Referee #3 · 7 Mar 2020

General comments: This paper investigated the long term trend and seasonal variation of total column ozone (TCO) and total ozone low over Tibetan Plateau (TP) by using the regression analysis. The impacts of individual variables including solar cycle, QBO, and geopotential height (GH) have been discussed. They found that the GH may play an important role influencing the TCO especially on 150hPa levels. Moreover, they mentioned there might be the dynamical controlling of Inter Tropical Convergence Zone, ENSO events or Walker circulation in the lower stratosphere. In this paper, the scientific conclusions may need to be addressed more carefully and clarified. Some other details could be found beneath in the "specific comments". I would also suggest the author to work on the writing of the manuscript.

Specific comments: 1. TOL has been discussed in the manuscript but it didn't show in

the abstract? 2. It would be more reasonable to address some discussion of reasons for choosing 4 TP regions. 3. Does Fig.1 show results of C3S? Some details would be helpful. 4. Fig.6, in QBO analysis, do purple dots represent combined QBO at 30hPa and 10hPa? 5. In the manuscript, SLIMCAT results show much smaller 150hPa GH contribution in DJF due to coarser resolutions. If it's correct, simulations with a finer resolution might be suggested to perform here. The values in JJA almost double in model simulations. It might require some discussions.

---

## Author Comment (AC1) · 18 Apr 2020

**Response to Reviewers' Comments.**

**Reviewer 1**

**1 Overall Remarks**
**The manuscript investigates interannual, as well as long-term variations of total ozone columns over the Tibetan Plateau and neighbouring regions. Analysis is based on the Copernicus Climate Change Service dataset, and on SLIMCAT**
**chemical transport model simulations. Direct observational data are not used. Main analysis tool is multiple linear regression. In principle this could provide some new and useful information to ACP readers. However, I found it very difficult to grasp meaningful main messages from the manuscript. In my opinion the manuscript presents a largely un-organized smorgas-board of regression**
**results, which may or may not be statistically significant. Results are taken largely at face-value. I did not see any clear scientific questions to be addressed, nor any stringent logical arguments towards answering such questions. In a similar fashion, the manuscript does cite a large number of papers, but I never see any coherent line-of-thought, how the present results would add new**
**knowledge to what is already out there.**
**In my opinion the manuscript needs a major rewrite and re-organization. The authors should first decide on their new and major results and then work out these main messages from their analysis. A much more clear and concise presentation is necessary. Right now, I feel that the most appropriate title for the**
**manuscript would be "Regressions that we did, and correlations that we found, for total ozone data near the Tibetan Plateau". This might be OK for a bachelor-thesis, but certainly not for an ACP paper.**

We thank the reviewer for the thoughtful comments and suggestions. We have made
substantial modifications to improve the quality of the paper. Our replies are given below with a description of what we have changed in the revised manuscript.

Our modifications include:
(1) Focus on the main points and make the paper clearer and more concise.
(2) Improve discussion of statistical significance.
(3) Adjust the structure of the paper.
(4) Delete or remove some figures and discussions.
(5) Update the results with the updated discussions.
(6) Add observations to validate the C3S ozone and ECMWF GH data.
(7) Recheck the references.

Three main points based on the major results are listed as follows:
- The Tibetan Plateau (TP) is showing asymmetrical (slower) ozone recovery compared to the zonal mean over the same latitude band.

• The 150 hPa geopotential height (GH150) is a more realistic dynamical proxy (than previously used surface temperature) for TP column ozone. It influences summertime TCO variations over the TP through persistence of the wintertime ozone signal.
• Model results confirm that wintertime TP ozone variations are largely controlled by tropics-to-high latitude transport processes whereas summertime concentrations are
combined effect of photochemical decay and tropical processes.

Based on the updated main points, we rename our manuscript from "Decreases in wintertime total column ozone over the Tibetan Plateau during 1979-2017" to "Analysis and attribution of total column ozone changes over the Tibetan Plateau during 1979-2017". We have revised
the abstract and the conclusions to be a clearer reflection of the main points and major results. Please see lines 17-47 and lines 529-563 in the revised manuscript.

**2 Detailed Suggestions**
**Abstract: EESC says TP ozone should go up since 1997, PWLT and OLR say ozone is going down or staying constant. Does this difference mean anything? Is it even statistically significant? Is EESC even a good / relevant proxy for TP ozone? Does the GH150 proxy explain some of the ozone trend? Is there a trend in the GH150 proxy (or in surface temperature)? Is that relevant for the ozone**
**trend? Do the SLIMCAT experiments provide more information about the underlying processes than the C3S data? The fact that regression of SLIMCAT data gives very similar results to regression of C3S data does not provide new insight. Both should give very similar results, because underlying meteorological conditions are very similar, and SLIMCAT should represent chemistry**
**reasonably well. Does the comparison between 2004 and 2008 in SLIMCAT data provide anything different from comparing 2004 and 2008 in C3S data? All these questions come up when reading abstract and paper. None of them is answered satisfactorily.**

Reply: We thank the reviewer for the comments and agree that some of the conclusions in the previous version were confusing. Also, we apologise that we found a bug in our processing routine of the seasonal means (the last DJF value in 2017 should include the Jan and Feb data in 2018), hence our new results are slightly different. The updated results and discussions are shown and marked in red in the revised manuscript. Based on the updated major results and
the three main points, we have revised our abstract (Please see lines 17-47). We feel that these address the above general points.

**(1) EESC says TP ozone should go up since 1997, PWLT and OLR say ozone is**
**going down or staying constant. Does this difference mean anything? Is it even statistically significant? Is EESC even a good / relevant proxy for TP ozone?**

Reply: In the updated abstract, we present the methods and results for the long-term ozone trend analysis: "We use piecewise linear trend (PWLT) and equivalent effective stratospheric chlorine loading (EESC)-based multi-variate regression models with various proxies to attribute the influence of dynamical and chemical processes on the TCO variability." and "Both regression models show that the TP column ozone trends change from negative trends from 1979-1996 to small positive trends from 1997-2017, although the later positive trend based on PWLT is not statistically significant. The wintertime positive trend since 1997 is larger than that in summer, but both seasonal TP recovery rates are smaller than the zonal means over the same latitude band."

We also expand Section 4.2 where we use PWLT and EESC-based multi-variate regression models to study the long-term ozone trends and variability over the TP. The ordinary least square regression (OLR) method in the old version has been removed to avoid confusion. Both remaining regression models show that TCO trends over the TP decrease during 1979-1996 and go up since 1997. The comparison between the PWLT and EESC-based trends (Table 3 in Section 4.2 and Table S3 in the Supplement) shows a good agreement, except that EESC trends are statistically significant within 2σ for the full data record. However, the positive trends in PWLT are always not significant (please see the description in lines 353-362). That is, EESC is a good and relevant proxy for the TP ozone variability, which represents the chemical contribution to the ozone trend.

**(2) Does the GH150 proxy explain some of the ozone trend? Is there a trend in the GH150 proxy (or in surface temperature)? Is that relevant for the ozone trend?**

Reply: In the updated abstract, we present the regression results and the discussion concerning GH150: "For TP column ozone, both regression models suggest that the geopotential height at 150 hPa (GH150) is a more suitable and realistic dynamical proxy compared to a surface temperature proxy used in some previous studies. Our analysis also shows that the wintertime GH150 plays an important role in determining summertime TCO over TP through persistence of the ozone signal."

The information about GH150 proxy (or the surface temperature) is given in Section 4.1. The GH150 proxy is relevant to the ozone trend (variations). Table 1 shows that the correlation between the TCO and GH150 is -0.514 (above the 99% confidence level). Figure S3 in the Supplement shows an increasing trend in the GH150 proxy. However, in our regression, the GH150 proxy is de-trended so is independent of other trend proxies (EESC or PWLT). The regression analysis in Section 4.2 indicates that GH150 is a more suitable and realistic dynamical proxy for the wintertime ozone variability over the TP compared to the surface temperature. The significant correlation between wintertime and summertime highlights the dynamical influence of the wintertime GH150 on the summertime ozone concentrations.

**(3) Do the SLIMCAT experiments provide more information about the underlying processes than the C3S data? The fact that regression of SLIMCAT data gives very similar results to regression of C3S data does not provide new insight. Both should give very similar results, because underlying meteorological conditions are very similar, and SLIMCAT should represent chemistry reasonably well.**

Reply: Yes, we do believe that the SLIMCAT experiments provide additional information about the underlying processes beyond the C3S data. We use SLIMCAT sensitivity experiments to explore the role that GH150 plays in the DJF mean ozone variations over the TP region (as described in Section 5). This information cannot be obtained from the C3S data.

It is true that the regression of SLIMCAT control experiment output gives very similar results to regression of C3S data with very similar underlying meteorological conditions. Therefore, in the revised abstract we have omitted the content about the SLIMCAT control experiment. In the revised manuscript (Section 5), we also have moved the figures about the SLIMCAT control experiment results to the Supplement (Figures S5 and S6), mainly focusing on the discussion of the sensitivity experiments and the results.

**(4) Does the comparison between 2004 and 2008 in SLIMCAT data provide anything different from comparing 2004 and 2008 in C3S data?**

Reply: Basically, SLIMCAT is used for time-slice-type simulations with fixed dynamical conditions so that effect of short-term variability on ozone (e.g. time varying QBO, ENSO, solar fluxes, ODS changes) are avoided for a particular year. A 5-year mean comparison between 2004 and 2008 in SLIMCAT does provide some additional insight about "dynamics-only" influence on the ozone change compared to just comparing C3S data for 2004 and 2008. This is very important in the lower stratosphere where the ozone lifetime is of the order of years. We don't think that the comparison between 2004 and 2008 in SLIMCAT data and C3S data would be enough to elucidate the role GH at 150 hPa plays over the TP.

Thus, in the abstract we introduce the SLIMCAT sensitivity experiments and discuss the simulation results: "We also use a 3-D chemical transport model to diagnose the contributions of different proxies for the TP region. The role of GH150 variability is illustrated by using two sensitivity experiments with repeating dynamics of 2004 and 2008. Simulated ozone profiles clearly show that wintertime TP ozone concentrations are largely controlled by tropics to mid-latitude pathways, whereas in summer variations associated with tropical processes play an important role." In the revised manuscript, we also have added some information to make it clearer (lines 213-215 in Section 2.2 and lines 459-463 in Section 5).

**Section 2.1: C3S is based on model assimilation of meteorological and ozone observations. It would be very important to check if the results in the paper are**

**also valid for real ozone observations (e.g. from the SBUV or GOME series of satellites, or, even better, from a nearby total ozone station) and for real geopotential heights, e.g. from a nearby radiosonde station. Such data should be**
**added to the manuscript / plots.**

Reply: We thank the reviewer for the suggestion. In the revised manuscript, we use the direct ozone observations from the SBUV series of satellites to validate the results based on C3S. The information about SBUV is briefly introduced in Section 2.1 (lines 155-159). In Section 3,
C3S-based total ozone data and SBUV satellite-based observations over the North-TP latitude band (40°-50°N) and the South-TP latitude band (10°-20°N) are compared and shown in Figures 1 (a) and (b). As SBUV total columns are assimilated in C3S, their differences are less than 2-3% throughout the entire data record (as shown in the supplementary Figure S1).

To check if the 150 hPa geopotential height (GH150) data from ECMWF are realistic, we make the comparison of the DJF mean GH150 from ECMWF ERA-Interim with those from a nearby radiosonde station (http://weather.uwyo.edu/upperair/seasia.html). As shown in Section 4.1 (lines 292-295) as well as the supplementary Figure S3, both ECMWF and radiosonde GH150 data averaged in DJF over Lhasa (30°N, 91°E) show an increasing trend during the
time period 1979-2017 with a statistically significant correlation (0.96). However, as mentioned earlier GH150 and ST proxies are de-trended before their inclusion in the regression model, hence they represent only inter-annual dynamical variability.

**Section 2.2: Are the SLIMCAT model results really important for the main messages of the paper? If not, maybe just omit them. To me, it never becomes clear what additional insight comes from the (coarser resolution than C3S) SLIMCAT simulations here.**

Reply: In the revised manuscript, we clarify how SLIMCAT sensitivity experiments help to elucidate the role of GH150 on the variability. The composite analysis in Section 5 demonstrates that seasonal fluctuations in GH150 play a key role in controlling the DJF mean TCO variability over the TP, which may be associated with ITCZ, ENSO events or Walker circulation.

To make the paper clearer and more concise, we have reduced information about the SLIMCAT control simulations and added it to the Supplement (Figures S5 and S6). We also have tried to improve the description about the SLIMCAT sensitivity simulations in Section 5 (lines 447-463).

**Section 2.3: I found this section, especially the results part, out-of-place and confusing here. To me, a more logical flow would be to present the total ozone time-series first, then, later, the multiple linear regression and its results**

**Beginning of Section 4: That would be the right place for the description of the MLR /Equation 2 and for the presentation of Table 1.**

Reply: We thank the reviewer for the suggestion. We have rearranged the structure of the paper. Section 2.3 with the description of the regression models is moved to Section 4.1. The analysis results based on the regression models are now presented in Section 4.2.

The new structure of the revised manuscript is described in the lines 141-146: "The layout of the paper is as follows. Section 2 introduces the C3S ozone dataset and TOMCAT/SLIMCAT model used for the analysis of the total ozone variability. The long-term TCO time series and TOL over the TP region are presented in Section 3. Regression methods as well as analysis of the contribution of different proxies to the total ozone variations in different seasons are given in Section 4. Section 5 discusses sensitivity experiment results based on TOMCAT/SLIMCAT and is followed by our summary and conclusions in Section 6."

**Figure 3 (especially 3b) and Figure 4: How relevant are these for the main new messages of the paper? Is there much value in a simple linear trend over the entire period? Especially, using and discussing this simple linear trend, to me, creates confusion and mis-understanding for the later use of the more comprehensive regression (which includes the additional proxies). I would suggest to omit both figures and their corresponding text.**

Reply: We thank the reviewer for the comments. To avoid confusion with the later trend analysis using multi-variate linear regression models, we have omitted the old Figure 3 (b) and Figure 4 and their corresponding text. As the seasonal variability in old Figure 3 (a) is associated with the TOL over the TP (Section 3) and contributions of different proxies in different seasons (Section 4.2), we have removed it to the supplementary Figure S2.

**Figure 10: How would that Figure look for C3S data? How would it look for a single model year? I would assume that each SLIMCAT year would be very similar for repeating meteorology. Discussion of Fig. 10: From Fig. 10 it is not clear to me what causes what. Especially in JJA, there are really no distinctive features near the TP. If there are signatures of circulation cells, e.g. with locally high / low ozone, these circulation cells (Walker, ENSO) should be indicated (e.g. by arrows) in the figure. As it is now, the discussion of Fig. 10 is largely speculative / hand-waving. Not very convincing to me.**

Reply: As we have mentioned above, the comparison between 2004 and 2008 in C3S data does not make any sense for elucidating the role GH150 plays in the ozone variability over the TP, because it is impossible to control most dynamical and chemical factors to be similar and only keep the factor GH150 as the major contributor to the C3S ozone variations.

Neither does it make sense to look for a single model year, even though each SLIMCAT year
       would be very similar for repeating meteorology. The reason we use the 5-year averaged
       SLIMCAT sensitivity experiments is to exclude the influence from short-term processes (e.g.
       aerosol, solar cycle) and mainly study the contribution from the dynamical GH150 proxy. If
       we only take a single model year for sensitivity experiments, the ozone anomalies will be
different from those averaged for 5 years (Figure 6 in Section 5), because at this time not only
       GH150 but also other chemical proxies have contributed to the ozone variations in the year
       2004 and 2008. Due to the long ozone lifetime their influences are indistinguishable so that
       we cannot judge the role GH150 plays in the ozone variations over the TP.

Discussion of Figure 10 (the new Figure 6 in the revised manuscript) is to indicate and
       explain why the wintertime ozone difference over the TP is caused by the GH150 difference.
       In JJA, there are no distinctive features near the TP. For the years 2004 and 2008, the
       distinctive GH150 difference happens in DJF not in JJA, so we mainly discuss the GH150
       role in the wintertime ozone differences over the TP. In the revised version, we have tried to
improve and clarify the discussion (lines 477-484 and 500-514). The model results support the
       hypothesis that wintertime TP ozone variations are largely controlled by tropics-to-high
       latitude transport processes whereas summertime concentrations are combined effect of
       photochemical decay and tropical processes (lines 511-514). In the new Figure 6, we have
       added arrows in (a) and (c) to indicate the GH differences influenced by those from the high
and low latitudes; the dashed blue and red boxes in (b) are also marked to indicate the
       negative and positive ozone anomalies over the TP and the Pacific Ocean. Based on the
       composite analysis, we conclude that "GH150 fluctuations play a key role in controlling the
       DJF mean TCO variability over the TP, which may be associated with ITCZ, ENSO events or
       Walker circulation." (lines 555-557).

       **Table 1, 2. Are these correlations and regressions taken for the individual months
       D, J, F, or for the 3-month DJF average. From the figures it seems like the latter
       is the case. However, this should clarified in the table captions.**

       Reply: The correlations and regressions in Table 1, 2 are taken for the 3-month DJF average.
       They are now clarified in the table captions in the revised manuscript (lines 295 and 340).

**Table 2 and its discussion: Increasing R or $R^2$ with the addition of new predictors
       is quite normal. Every additional predictor is likely to pick up some variance, but
       this may be random and meaningless. This might be the case for the slightly
       larger R for PWLT over EESC, because PWLT or 2-Linear-Trends add
       additional predictors. So the authors should be careful not to over-interpret
small changes in R. Try it, take any additional random time series, and add it as
       a predictor. It will increase the overall R. So the real question is: Is the increase
       in R significant? There are tests for this (e.g. F-test, or adjusted R). These should
       be used, and small changes in R should not be over-interpreted.**

Reply: We thank the reviewer for the comments. We agree with the reviewer that the coefficient of determination (R-squared) increases with the addition of new predictors, and the slightly larger R-squared for PWLT over EESC should not be over-interpreted in the paper. Therefore, we have deleted the R-squared comparison between the EESC and PWLT in Table 2.

In the revised manuscript, we have used adjusted R-squared to assess the overall significance of the regression models (Table 2 in Section 4.2 for the PWLT regression model and Table S2 in the Supplement for the EESC-based regression model). We also used t-test statistics with standard errors within 2σ to check the significance of each proxy in the regression model (Table 3 in Section 4.2 and Table S4 in the Supplement for PWLT and EESC-based regression models, respectively). Both regression models show that the wintertime GH150 proxy improves the adjusted determination coefficient for the TP region and is statistically significant above the 99% confidence level.

**Table 3: Are the stated uncertainties 1σ or 2σ? Please state. Even with 2σ, many of the discussed difference would, at the most, be borderline significant. With 1σ, uncertainties should be doubled, and nearly all statistical significance would disappear.**

**Table 4: Again, is the given uncertainty / std. err 1σ or 2σ?**

Reply: The uncertainties in the old Table 3 and Table 4 (also the new Table 3 in Section 4.2 and Table S3 in the Supplement) are 2σ. With 2σ uncertainties, most factors in these tables are statistically significant. We also now make it clear in these tables and the corresponding texts (lines 348-349 and 359-361).

**Fig. 8 and its discussion: Does this add anything substantial over what is already**

**known, e.g. from Fioletov and Shepherd (2003) or Holton and Tan (1980)? If not, does it add anything to the salient main messages of the paper? If not, maybe just drop it?**

Reply: Yes, we introduce the results from Fioletov and Shepherd (2003) and Holton and Tan (1980) with the purpose to further explore what happens to the TP region, e.g. if the transport in wintertime ozone buildup persists through the summer, if the QBO still dominates the transport which modulates the wintertime ozone buildup and persists through the summertime, and if there exist some different proxy contributions during the seasonal persistence.

In the Introduction part, we expanded this discussion (lines 127-133) as: "In different seasons, especially in winter and summer, the proxies affecting the ozone variations over the TP may also be different due to some complicated mechanisms. Fioletov and Shepherd (2003) have studied the seasonal persistence of mid-latitude total ozone anomalies and demonstrated that ozone values are correlated through the annual cycle from the buildup in winter-spring to the ozone minimum in autumn. For mid-latitudes, the tropical zonal winds as manifested in the QBO dominate the wintertime ozone buildup (Holton and Tan, 1980). However, such information remains unknown for the large local TP region."

In the revised manuscript, we also have improved the discussion of the old Figure 8 (the new

Figure 5 in Section 4.2) in lines 431-433: "Fioletov and Shepherd (2003) highlighted the seasonal persistence of mid-latitude total ozone anomalies and indicated that seasonal predictability is applicable for latitudinal belts or large regions only." and lines 435-443: "The ozone buildup in wintertime when transport dominates is largely modulated by QBO (Holtan and Tan, 1980); however, GH150 represents large part of wintertime variability in the ozone transport. In summertime, as expected, photochemical processes become more important while dynamical impact from QBO decreases and almost disappears for GH150. Seasonal persistence in TCO anomalies shows that if there is more transport in DJF as represented by GH150 changes, higher ozone values will persist for at least 6 months, even though there is little correlation between summertime ozone anomalies and GH150. This analysis clearly highlights dynamical influence of the wintertime GH150 on the summertime (JJA) ozone concentrations."

---

## Author Comment (AC2) · 18 Apr 2020

**Response to Reviewers' Comments.**

**Reviewer 3**

**General comments: This paper investigated the long term trend and seasonal variation of total column ozone (TCO) and total ozone low over Tibetan Plateau (TP) by using the regression analysis. The impacts of individual variables including solar cycle, QBO, and geopotential height (GH) have been discussed. They found that the GH may play an important role influencing the TCO especially on 150hPa levels. Moreover, they mentioned there might be the dynamical controlling of Inter Tropical Convergence Zone, ENSO events or Walker circulation in the lower stratosphere. In this paper, the scientific conclusions may need to be addressed more carefully and clarified. Some other details could be found beneath in the "specific comments". I would also suggest the author to work on the writing of the manuscript.**

We thank the reviewer for the helpful comments and suggestions. We have made substantial modifications to improve the quality of the paper. The three main points based on our major results are listed as follows:

- The Tibetan Plateau (TP) is showing asymmetrical (slower) ozone recovery compared to the zonal mean over the same latitude band.
- The 150 hPa geopotential height (GH150) is a more realistic dynamical proxy (than previously used surface temperature) for TP column ozone. It influences summertime TCO variations over the TP through persistence of the wintertime ozone signal.
- Model results confirm that wintertime TP ozone variations are largely controlled by tropics-to-high latitude transport processes whereas summertime concentrations are combined effect of photochemical decay and tropical processes.

Based on the updated main points, we rename our manuscript: "Analysis and attribution of total column ozone changes over the Tibetan Plateau during 1979-2017". The abstract and the conclusions are also revised based on the three main points and our updated major results.

Our replies to the reviewer's specific comments are given below with a description of what we have changed in the revised manuscript.

**1. Results of TOL in abstract?**

Reply: In our updated abstract, we have added some results of TOL in lines 25-33: "We also compare the seasonal behaviour of the relative total ozone low (TOL) over the TP with the zonal mean at the same latitude. Both regression models show that the TP column ozone trends change from negative trends from 1979-1996 to small positive trends from 1997-2017, although the later positive trend based on PWLT is not statistically significant. The wintertime positive trend since 1997 is larger than that in summer, but both seasonal TP recovery rates are smaller than the zonal means over the same latitude band."

**2. Some discussion of reasons for choosing 4 TP regions?**

Reply: In the revised manuscript (Section 2.1), we have added a discussion of reasons for choosing 4 TP regions (lines 184-187): "These regions represent the tropics and mid-latitudes with the TP and zonal TP in the critical zone. We choose them to compare the contribution of different dynamical proxies to their ozone variations, especially over the TP region."

**3. Fig.1 shows results of C3S?**

Reply: The new Figure 1 in the revised manuscript shows the TCO time series based on C3S and SBUV. As C3S is based on model assimilation of meteorological and ozone observations, we use the direct ozone observations from the SBUV series of satellites to validate the results based on C3S. Their differences are less than 2-3% throughout the data record and are shown in the supplementary Figure S1.

**4. Fig.6, in QBO analysis, purple dots represent combined QBO at 30hPa and 10hPa?**

Reply: Yes, purple dots in the old Figure 6 represent combined QBO at 30 hPa and 10 hPa. In the revised manuscript, we have re-plotted the new Figure 3 with updated plot legend to make it easier to understand.

**5. SLIMCAT results show much smaller 150 hPa GH contribution in DJF due to coarser resolutions? Simulations with a finer resolution might be suggested to perform here. The values in JJA almost double in model simulations. It might need some discussions.**

Reply: Actually, our updated results (with the early 2018 data included to determine the last DJF value in 2017) show that SLIMCAT simulation results are similar to the C3S regression results, although contributions from most explanatory proxies are larger except for the GH150 in DJF. This difference is probably due to the coarse model resolution and the inhomogeneities in ERA-Interim data (lines 451-454).

TOMCAT/SLIMCAT is a global 3D off-line chemistry-transport model widely used to study the processes controlling tracer distributions in the atmosphere. The resolution of the simulations in the paper is $2.8^{o} \times 2.8^{o}$, which is coarser than the resolution of C3S ($0.5^{o} \times 0.5^{o}$). That may be one reason why SLIMCAT results are different from those based on C3S.

It is true that the version of the model with higher resolution would be expected to present a more realistic representation of ozone. Feng et al. (2005) has indicated that SLIMCAT with higher resolution ($2.8^{o} \times 2.8^{o}$) shows more reasonable transport and mixing than the lower resolution ($7.5^{o} \times 7.5^{o}$). With higher resolution, chemical ozone depletion reproduced by the model is generally larger, which agrees better with observations. However, it should be noted that Feng et al. (2011) also investigated the effect of resolution in the CTM (from $5.6^{o} \times 5.6^{o}$ to

1.1° ×1.1°) on the convective mass fluxes and found that the changes are small. For polar stratospheric studies, Grooß et al. (2018) also found that there is not much difference in the time series of HCl which affects the simulated ozone depletion using two resolutions (1.2°×1.2° and 2.8°×2.8°) of TOMCAT/SLIMCAT. Hence, simulations with a finer resolution may not promise a large improvement on the 150 hPa GH contribution compared to the C3S results. Given the prohibitive computational cost of performing high resolution simulations, and the improvement in the presented results compared to the submitted version, we have kept the moderate resolution 2.8°×2.8° simulations.

Our simulation results show overestimated contributions from the different explanatory factors when compared to the C3S regression results in both DJF and JJA (the values in JJA almost double). Some differences are expected because there are uncertainties in the model simulations. The complex set of processes in the model (e.g. chemistry, photolysis, dynamics and emission) and the quality of meteorological analysis data used will inevitably cause the uncertainties in the model. Therefore, in the revised manuscript, we briefly summarise the control simulation results, and our focus is on the two relative sensitivity experiments to investigate the role of wintertime GH150 on ozone transport (see also response to Reviewer 1). The simulated ozone profiles clearly show that wintertime TP ozone concentrations are largely controlled by tropics-to-mid-latitude pathways, whereas in summer variations associated with tropical processes play an important role (lines 42-44 and Section 5).

**References:**

Feng, W., Chipperfield, M. P., Davies, S., Sen, B., Toon, G., Blavier, J. F., Webster, C. R., Volk, C. M., Ulanovsky, A., Ravegnani, F., von der Gathen, P., Jost, H., Richard, E. C., and Claude, H.: Three-dimensional model study of the Arctic ozone loss in 2002/2003 and comparison with 1999/2000 and 2003/2004, Atmos. Chem. Phys., 5, 139–152, https://doi.org/10.5194/acp-5-139-2005, 2005.

Feng, W., Chipperfield, M. P., Dhomse, S., Monge-Sanz, B. M., Yang, X., Zhang, K., and Ramonet, M.: Evaluation of cloud convection and tracer transport in a three-dimensional chemical transport model, Atmos. Chem. Phys., 11, 5783–5803, https://doi.org/10.5194/acp-11-5783-2011, 2011.

Grooß, J.-U., Müller, R., Spang, R., Tritscher, I., Wegner, T., Chipperfield, M. P., Feng, W., Kinnison, D. E., and Madronich, S.: On the discrepancy of HCl processing in the core of the wintertime polar vortices, Atmos. Chem. Phys., 18, 8647–8666, https://doi.org/10.5194/acp-18-8647-2018, 2018.

---

## Author Comment (AC3) · 18 Apr 2020

**By Qian LI**

**1. Is there any result or discussion of TOL in the abstract? 2. Please explain the reason why choosing 4 TP regions? 3. Does Fig.1 show results of C3S? 4. In Fig.6, for QBO analysis, please make sure whether purple dots represent combined QBO at 30hPa and 10hPa? 5. SLIMCAT results show much smaller 150hPa GH contribution in DJF due to coarser resolutions? Simulations with a finer resolution might be suggested to perform here. The values in JJA almost double in model simulations. It might need some discussions.**

We thank the reviewer for the helpful comments and suggestions. We have made substantial modifications to improve the quality of the paper. The three main points based on our major results are listed as follows:

- The Tibetan Plateau (TP) is showing asymmetrical (slower) ozone recovery compared to the zonal mean over the same latitude band.
- The 150 hPa geopotential height (GH150) is a more realistic dynamical proxy (than previously used surface temperature) for TP column ozone. It influences summertime TCO variations over the TP through persistence of the wintertime ozone signal.
- Model results confirm that wintertime TP ozone variations are largely controlled by tropics-to-high latitude transport processes whereas summertime concentrations are combined effect of photochemical decay and tropical processes.

Based on the updated main points, we rename our manuscript: "Analysis and attribution of total column ozone changes over the Tibetan Plateau during 1979-2017". The abstract and the conclusions are also revised based on the three main points and our updated major results.

Our replies to the reviewer's specific comments are given below with a description of what we have changed in the revised manuscript.

**1. Is there any result or discussion of TOL in the abstract?**

Reply: In our updated abstract, we have added some results of TOL in lines 25-33: "We also compare the seasonal behaviour of the relative total ozone low (TOL) over the TP with the zonal mean at the same latitude. Both regression models show that the TP column ozone trends change from negative trends from 1979-1996 to small positive trends from 1997-2017, although the later positive trend based on PWLT is not statistically significant. The wintertime positive trend since 1997 is larger than that in summer, but both seasonal TP recovery rates are smaller than the zonal means over the same latitude band."

**2. Please explain the reason why choosing 4 TP regions?**

Reply: In the revised manuscript (Section 2.1), we have added a discussion of reasons for choosing 4 TP regions (lines 184-187): "These regions represent the tropics and mid-latitudes with the TP and zonal TP in the critical zone. We choose them to compare the contribution of different dynamical proxies to their ozone variations, especially over the TP region."

**3. Does Fig.1 show results of C3S?**

Reply: The new Figure 1 in the revised manuscript shows the TCO time series based on C3S and SBUV. As C3S is based on model assimilation of meteorological and ozone observations, we use the direct ozone observations from the SBUV series of satellites to validate the results based on C3S. Their differences are less than 2-3% throughout the data record and are shown in the supplementary Figure S1.

**4. In Fig.6, for QBO analysis, please make sure whether purple dots represent combined QBO at 30hPa and 10hPa?**

Reply: Yes, purple dots in the old Figure 6 represent combined QBO at 30 hPa and 10 hPa. In the revised manuscript, we have re-plotted the new Figure 3 with updated plot legend to make it easier to understand.

**5. SLIMCAT results show much smaller 150hPa GH contribution in DJF due to coarser resolutions? Simulations with a finer resolution might be suggested to perform here. The values in JJA almost double in model simulations. It might need some discussions.**

Reply: Actually, our updated results (with the early 2018 data included to determine the last DJF value in 2017) show that SLIMCAT simulation results are similar to the C3S regression results, although contributions from most explanatory proxies are larger except for the GH150 in DJF. This difference is probably due to the coarse model resolution and the inhomogeneities in ERA-Interim data (lines 451-454).

TOMCAT/SLIMCAT is a global 3D off-line chemistry-transport model widely used to study the processes controlling tracer distributions in the atmosphere. The resolution of the simulations in the paper is $2.8^{o} \times 2.8^{o}$, which is coarser than the resolution of C3S ($0.5^{o} \times 0.5^{o}$). That may be one reason why SLIMCAT results are different from those based on C3S.

It is true that the version of the model with higher resolution would be expected to present a more realistic representation of ozone. Feng et al. (2005) has indicated that SLIMCAT with higher resolution ($2.8^{o} \times 2.8^{o}$) shows more reasonable transport and mixing than the lower resolution ($7.5^{o} \times 7.5^{o}$). With higher resolution, chemical ozone depletion reproduced by the model is generally larger, which agrees better with observations. However, it should be noted that Feng et al. (2011) also investigated the effect of resolution in the CTM (from $5.6^{o} \times 5.6^{o}$ to

1.1$^o$ $\times$1.1$^o$) on the convective mass fluxes and found that the changes are small. For polar stratospheric studies, Grooß et al. (2018) also found that there is not much difference in the time series of HCl which affects the simulated ozone depletion using two resolutions (1.2$^o$×1.2$^o$ and 2.8$^o$×2.8$^o$) of TOMCAT/SLIMCAT. Hence, simulations with a finer resolution may not promise a large improvement on the 150 hPa GH contribution compared to the C3S results. Given the prohibitive computational cost of performing high resolution simulations, and the improvement in the presented results compared to the submitted version, we have kept the moderate resolution 2.8$^o$×2.8$^o$ simulations.

Our simulation results show overestimated contributions from the different explanatory factors when compared to the C3S regression results in both DJF and JJA (the values in JJA almost double). Some differences are expected because there are uncertainties in the model simulations. The complex set of processes in the model (e.g. chemistry, photolysis, dynamics and emission) and the quality of meteorological analysis data used will inevitably cause the uncertainties in the model. Therefore, in the revised manuscript, we briefly summarise the control simulation results, and our focus is on the two relative sensitivity experiments to investigate the role of wintertime GH150 on ozone transport (see also response to Reviewer 1). The simulated ozone profiles clearly show that wintertime TP ozone concentrations are largely controlled by tropics-to-mid-latitude pathways, whereas in summer variations associated with tropical processes play an important role (lines 42-44 and Section 5).

**References:**

Feng, W., Chipperfield, M. P., Davies, S., Sen, B., Toon, G., Blavier, J. F., Webster, C. R., Volk, C. M., Ulanovsky, A., Ravegnani, F., von der Gathen, P., Jost, H., Richard, E. C., and Claude, H.: Three-dimensional model study of the Arctic ozone loss in 2002/2003 and comparison with 1999/2000 and 2003/2004, Atmos. Chem. Phys., 5, 139–152, https://doi.org/10.5194/acp-5-139-2005, 2005.

Feng, W., Chipperfield, M. P., Dhomse, S., Monge-Sanz, B. M., Yang, X., Zhang, K., and Ramonet, M.: Evaluation of cloud convection and tracer transport in a three-dimensional chemical transport model, Atmos. Chem. Phys., 11, 5783–5803, https://doi.org/10.5194/acp-11-5783-2011, 2011.

Grooß, J.-U., Müller, R., Spang, R., Tritscher, I., Wegner, T., Chipperfield, M. P., Feng, W., Kinnison, D. E., and Madronich, S.: On the discrepancy of HCl processing in the core of the wintertime polar vortices, Atmos. Chem. Phys., 18, 8647–8666, https://doi.org/10.5194/acp-18-8647-2018, 2018.

---

## Referee Report (RR1)

The Authors have revised the manuscript thoroughly and carefully based on all the comments indicated by the referee. So far, the referee is very satisfied with the revised manuscript. It is recommended to publish the manuscript as it is.

---

## Author Response (AR2)

**Response to the reviewer's comments on the minor revisions:**

**The authors have successfully addressed nearly all suggestions brought up by the reviewers. The manuscript has improved substantially and is now more or less ready for publication.**

**I do have a few more suggestions, though:**

**1.) The Introduction is still too long and too wordy. Many of the mentioned things will be well known to ACP readers. I strongly suggest to shorten the Introduction to about 50% (from 96 lines to about 50 lines), and to drop papers that do not need to be cited.**

Reply: We thank the reviewer for the comments and suggestions. We have rewritten the Introduction and shortened it to about 50 lines (as shown in lines 50-101 in the revised manuscript). The detailed modifications with track changes are shown in the marked-up manuscript attached behind.

**lines 178 to 181: The two cited papers deal with ozone profiles, NOT total ozone columns. I don't think they should be cited here, and I suggest to drop the sentence and the two references.**

Reply: The sentence and the two references are dropped in the revised manuscript (line 134).

**line 303: "fitting coefficients" should be "fitting results". The coefficients will change, but not the results.**

Reply: "fitting coefficients" is changed to "fitting results" in the revised manuscript (line 257).

**line 475: "is independent" should be "are independent"**

Reply: "is independent" is changed to "are independent" in the revised manuscript (line 429).

**line 563: "needs" -> "need"**

Reply: We keep the original "needs" in the revised manuscript (line 517) because "the ozone column …. needs monitoring" is correct English.

[revised manuscript text omitted]